# Mapping the structure-function relationship along macroscale gradients in the human brain

Evan Collins [1,2,3,4] ✉, Omar Chishti [1,2,5], Sami Obaid[1,6,7], Hari McGrath [1,8], Alex King[1,9], Xilin Shen[10], Jagriti Arora[10], Xenophon Papademetris[2,10,11], R. Todd Constable [1,2,10,12], Dennis D. Spencer[1] & Hitten P. Zaveri [13]

Functional coactivation between human brain regions is partly explained by white matter connections; however, how the structure-function relationship varies by function remains unclear. Here, we reference large data repositories to compute maps of structure-function correspondence across hundreds of specific functions and brain regions. We use natural language processing to accurately predict structure-function correspondence for specific functions and to identify macroscale gradients across the brain that correlate with structure-function correspondence as well as cortical thickness. Our findings suggest structure-function correspondence unfolds along a sensory-fugal organizational axis, with higher correspondence in primary sensory and motor cortex for perceptual and motor functions, and lower correspondence in association cortex for cognitive functions. Our study bridges neuroscience and natural language to describe how structure-function coupling varies by region and function in the brain, offering insight into the diversity and evolution of neural network properties.

The relationship between structure and function drives the organization of many biological systems. For example, proteins adopt conformational structures which shape their biological functions. Analogously, structure is thought to affect function in the brain[1]. Complex networks of synapses enable signaling between neurons, forming neural circuits[2]. The tracts of axons that make up neural circuits constitute the white matter (WM) connectome, which enables both short- and long-distance interactions between brain regions[3,4]. These neuronal interactions are considered to be the physiological origin of meaningful behavior, perception, and cognition[5], hence,

suggesting structure-function (SF) coupling. How this observed SF correspondence varies for specific functions is incompletely understood[1].

With advancements in imaging technology, SF correspondence of the human brain has been computed using a variety of statistical and biophysical approaches[6-10]. To analyze SF correspondence means to assess the extent to which functional properties shared between brain regions can be attributed to their shared structural networks. This commonly involves testing for the relationship between structural connectivity (SC), i.e., the extent of white matter connections between

[1]Department of Neurosurgery, Yale School of Medicine, New Haven, CT, USA. [2]Department of Biomedical Engineering, Yale University, New Haven, CT, USA. [3]Department of Biological Engineering, Massachusetts Institute of Technology, Cambridge, MA, USA. [4]David H. Koch Institute for Integrative Cancer Research, Massachusetts Institute of Technology, Cambridge, MA, USA. [5]Max Planck School of Cognition, Leipzig, Germany. [6]Division of Neurosurgery, Department of Surgery, Faculty of Medicine, University of Montreal, Montreal, Quebec, Canada. [7]Neurosurgery Service, University of Montreal Hospital Center (CHUM), Montreal, Quebec, Canada. [8]Department of Clinical Neurosciences, University of Cambridge, Cambridge, UK. [9]Department of Plant and Microbial Biology, University of California, Berkeley, Berkeley, CA, USA. [10]Department of Radiology and Biomedical Imaging, Yale School of Medicine, New Haven, CT, USA. [11]Department of Biomedical Informatics and Data Science, Yale School of Medicine, New Haven, CT, USA. [12]Interdepartmental Neuroscience Program, Yale University, New Haven, CT, USA. [13]Department of Neurology, Yale School of Medicine, New Haven, CT, USA. ✉e-mail: evanc@mit.edu

pairs of brain regions, and functional connectivity (FC), i.e., the extent of systematic coactivation between pairs of brain regions.

Early studies computed global SF correspondence by calculating the correlation between whole-brain SC and whole-brain FC, making the limiting assumption that the factors driving SF correspondence are uniform across the brain[1,11]. Although SC and FC are significantly correlated in global measures, the majority of variance of these measures was unexplained, suggesting an imperfect correlation[1,11]. Recent research has shown SF correspondence varies across the cortex[9,10]. A study in 2019 with 40 subjects used a multiple linear regression model to evaluate whether SC from diffusion MRI (dMRI) predicts FC from resting-state fMRI (rsfMRI)[10]. The study demonstrated that SF correlations were higher in sensory-motor regions and lower in default mode network, following a cortical gradient they characterize as unimodal to transmodal[12,13]. Another study in 2019 suggested SF correspondence was higher for regions involved in motor activity[14]; however, how SF correspondence varies across many functions remains unexplored.

Hence, we identified two key areas for further investigation. First, most prior studies use imaging data from a relatively small cohort of subjects. Although having a small cohort from a single imaging facility helps minimize confounding variables, interindividual variability in both FC and SC data reduces statistical power, potentially leading to unrepresentative group-wise associations. Second, prior studies typically estimate FC from rsfMRI. Coactivation patterns at rest, however, are significantly different from those when performing tasks[15], so SF correspondence computed from rsfMRI may be different than that computed from task-based fMRI. Moreover, it is arguably more biologically and evolutionarily meaningful to evaluate how structure corresponds to function during specific mental tasks.

Altogether, the effect of task activation on SF correspondence is incompletely understood[1,14], and, more generally, a representative map of the SF relationship in the human brain across specific regions and specific functions is critically lacking. We sought to overcome these limitations by using big data and natural language processing (NLP) approaches to compute the most generalizable, representative map of the varying relationship between structure and function across the human brain. We sourced structural and functional data from large repositories and localized them to a high-resolution anatomical atlas. Structural data was acquired from the WM connectomes of 1065 subjects from the dMRI data of Human Connectome Project (HCP). Functional data was acquired from Neurosynth and NeuroQuery meta-analytic repositories of fMRI activation data across 334 different functions. We computed SF correspondence globally, locally, and by specific function. We next used NLP techniques to predict SF correspondence for specific functions and to identify macroscale gradients that correlate with structure-function correspondence across the brain. We observed SF correspondence varies along a sensory-fugal organizational axis, with higher correspondence in primary sensory and motor cortex for externally-oriented, perceptual functions and lower correspondence in association cortex for internally-oriented, cognitive functions. We lastly related SF correspondence to cortical thickness.

## Results

### Generating representative structural connectivities and functional connectivities

We sourced structural data in the form of streamline counts and lengths from the WM tractograms of $N = 1065$ healthy adult individuals from HCP (Fig. 1a). End-to-end streamline count matrices and streamline length matrices were generated for each individual using the Yale Brain Atlas parcellation. Yale Brain Atlas is a high-resolution anatomical atlas consisting of 696 parcels covering the cortex and hippocampus, amygdala, and corpus callosum[16]. In addition to its relevance for epilepsy studies[16,17], the Yale Brain Atlas parcellation was

well-suited for this study due to its relatively small and uniform parcel size, enabling precise and meaningful analyses of structural and functional patterns. Arithmetic averages of these matrices were computed to generate the group-level streamline count (SC-Count) and streamline length (SC-Length) matrices. In addition, we computed the pairwise Euclidean distances between centroids of Yale Brain Atlas parcels (SC-Eucl Dist) (Fig. 1b). These three group-level SC matrices were used as inputs for network analysis scripts from Zamani Esfahlani et al.[9] to generate additional group-level SC matrices, bringing the total of SC metrics considered to thirteen (Fig. 1c).

We sourced functional data from both Neurosynth and NeuroQuery meta-analytic repositories of fMRI activation data. Neurosynth is a large open-source platform that uses text-mining, NLP techniques, and meta-analysis to collate fMRI activation data from 14,371 peer-reviewed articles and provide a voxelwise map of activations for 1334 terms[18]. These terms in Neurosynth are not strictly tasks conventionally used for fMRI, as they emerge from NLP topic analysis. Nevertheless, the terms represent the major topics of the web-scraped articles, which include both rsfMRI- and diverse task-based studies. Many terms are basic positional descriptors of the voxel (e.g., "prefrontal"); therefore, we selected and synthesized a subset of 334 terms that describe functional tasks (e.g., "vision"). NeuroQuery was created more recently to update Neurosynth namely through a larger, updated dataset of text and the use of semantic smoothing based on word embeddings to merge the results of similar terms[19]. We additionally analyzed the activation data of NeuroQuery using the same 334 functional terms that we subsetted from Neurosynth.

Separately for Neurosynth and NeuroQuery, we extracted the z-score activation values for 334 functional terms across all voxels in MNI152 space (Fig. 1d). Activation values from constituent voxels were averaged for each parcel of the Yale Brain Atlas (Fig. 1e). To transform these parcel-wise average activation values into connectivity data, we computed cosine similarity between the activation vectors of each pair of parcels (Fig. 1f). In addition, to compare our meta-analytic FC approach with the rsfMRI-derived FC approach of prior studies[9,10], we computed a group-average rsfMRI connectivity matrix for 34 healthy age- and sex-representative subjects. Once the SC and FC measures were estimated, we proceeded to analyze SF relationships across 334 functions and 696 parcels (Fig. 1g).

### Structural and functional connectivities exhibit imperfect global correspondence

To compare with prior approaches, we first sought to analyze global SF correspondence across the whole brain, which for our analyses included the cortex and hippocampus, amygdala, and corpus callosum. Spectral clustering and diffusion mapping of parcels to visualize the different patterns of SC and FC illustrate inexact global correspondence (Figs. 2b, c, S1a; see "Methods"). From pairwise linear regression models between different SC and FC types across the whole brain, we observe $R^2$ values no greater than 0.3 (Fig. 2d) and Spearman $r$ absolute values no greater than 0.5 (Fig. S1b), again suggesting limited explained correspondence between structure and function globally. We additionally evaluated mean out-of-sample $R^2$ values across ten cross-validation folds to assess more appropriately the extent to which different SC types can predict FC (Fig. S1c); minimal differences exist between the in-sample $R^2$ values (Fig. 2d) and out-of-sample $R^2$ values (Fig. S1c).

To select optimal multiple linear regression models to predict either FC-Neurosynth, FC-NeuroQuery, or FC-rsfMRI, we plotted mean out-of-sample adjusted $R^2$ values and mean squared error (MSE) values across ten cross-validation folds with varying numbers of SC predictors (Fig. 2e). This prediction-centric approach was chosen to prevent overfitting, which can be the case when in-sample $R^2$ values are used for model selection (Fig. S1d). For each cross-validation fold and for each number of SC predictors, the optimal model plotted was

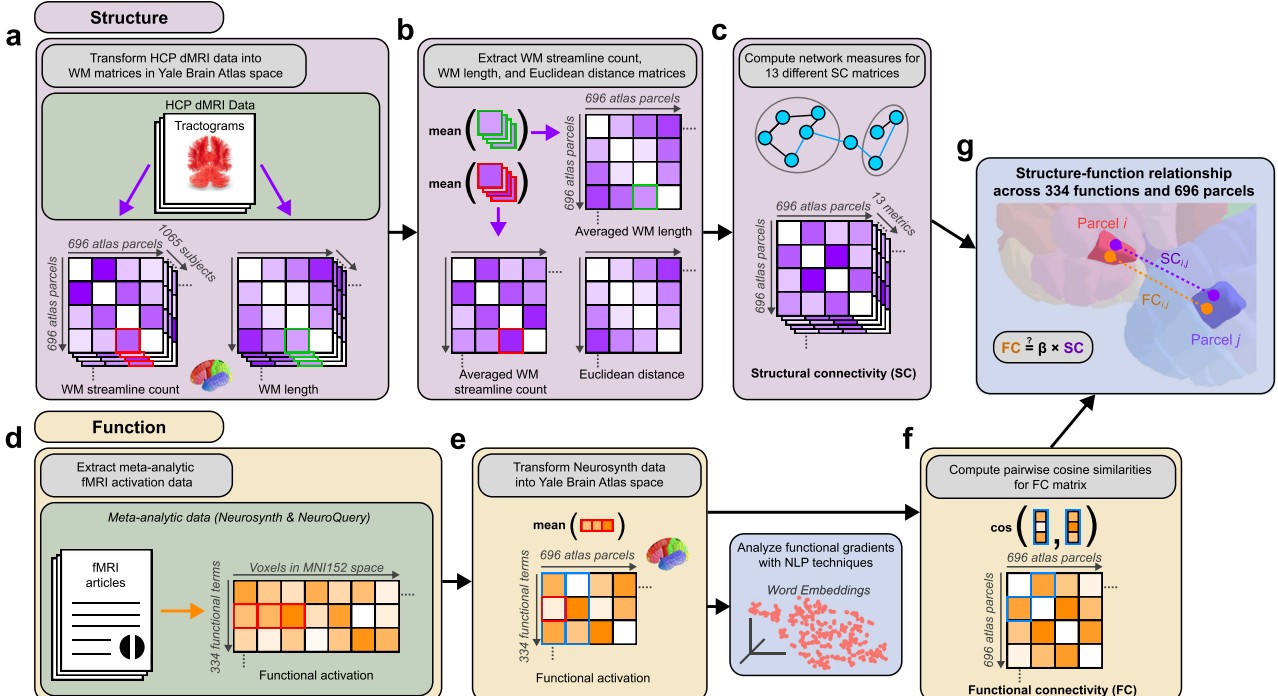

**Fig. 1 | Approach to deriving structural and functional connectivity data.**
**a** Structural measures of white matter (WM) streamline counts and WM lengths were acquired from the tractograms of $N = 1065$ healthy subjects from HCP and registered into Yale Brain Atlas space consisting of 696 parcels. **b** The averaged WM streamline count matrix, averaged WM length matrix, and Euclidean distance matrix were computed to create the aggregated SC matrices used as inputs for additional network analyses. **c** Network measures were computed from the three SC matrices of Fig. 1b to generate 13 total SC matrices, spanning a diversity of network measures to capture different characteristics of neuroanatomical organization. **d** For both Neurosynth and NeuroQuery, z-score activation values for 334

functional terms across all voxels in MNI152 space were extracted from the respective meta-analytic fMRI data repository. **e** Activation values from constituent voxels were averaged for each Yale Brain Atlas parcel. Separately, NLP techniques were employed for the 334 functional terms to analyze macroscale gradients. **f** Cosine similarity was lastly computed between every pair of parcels, in each instance comparing two vectors containing activation values across the 334 functional terms. The output is the functional connectivity (FC) matrix. **g** For each pair of parcels, we assessed the relationship between FC derived from Fig. 1f and the 13 SC measures derived from Fig. 1c. The result is the structure-function relationship across 334 functions and 696 parcels in the human brain.

selected via exhaustive search. For each FC type, we selected the optimal SC predictors that maximized out-of-sample adjusted $R^2$, where additional SC predictors did not significantly improve performance. The optimal SC predictors were cosine, flow graph, and navigation length for FC-Neurosynth; cosine and Euclidean distance for FC-NeuroQuery; and cosine, Euclidean distance, and flow graph for FC-rsfMRI (Fig. 2e). These optimal multiple linear regression models still have adjusted $R^2$ values no larger than 0.34, meaning the vast majority of functional variance is not explained by structural metrics globally. Altogether, this limited global SF correspondence motivated us to explore its variation by specific region and function.

**Structure-function correspondence is higher in primary sensory and motor cortex and lower in association cortex, and changes with task activation**
To evaluate how SF correspondence varies locally, we computed the strength of SF correspondence in each parcel as the adjusted $R^2$ value resulting from a multiple linear regression model relating FC with the optimal SC measures selected in Fig. 2e. This analysis was done for FC-Neurosynth (Fig. 3a), FC-NeuroQuery (Fig. 3b), and FC-rsfMRI (Fig. 3c). In addition, to enable effective comparisons between the SF correspondence maps produced from these three FC types, we plotted the $R^2$ by brain region, i.e., larger groupings of parcels (Figs. 3d, S4).

Overall, for all three FC types, we observed SF correspondence was higher in primary sensory and motor cortex closer to the longitudinal fissure, and lower in association cortex, aligning with a so-called sensory-fugal organizational axis[20,21]. Moreover, our SF correspondence gradients for FC-Neurosynth, FC-NeuroQuery, and

FC-rsfMRI largely align with that of a prior study which exclusively used FC-rsfMRI from a small cohort[10]. Comparing between our FC types, FC-Neurosynth and FC-rsfMRI exhibit $R^2 = 0.32$, FC-NeuroQuery and FC-rsfMRI $R^2 = 0.34$, and FC-Neurosynth and FC-NeuroQuery $R^2 = 0.37$. This overall alignment mitigated some of our concerns about the noise level from the meta-analytic approaches and motivated us to leverage the main advantage of these meta-analytic repositories, i.e., being able to analyze activity across hundreds of specific functions, in the next section of our study.

Nevertheless, we found using meta-analytic FC produced some significant differences compared to using FC-rsfMRI (Fig. 3d). When compared to FC-rsfMRI, using FC-Neurosynth yielded an even greater discrepancy in SF correspondence between primary sensory & motor cortex and association cortex. Using FC-NeuroQuery increased SF correspondence in many regions, most notably the orbitofrontal cortex. In addition, another marked difference from FC-rsfMRI was observed in the amygdala, with significantly greater SF correspondence when using either meta-analytic FC. These findings underscore the impact of using meta-analytic data, i.e., diverse fMRI data across many tasks and states, in SF analysis.

We next considered the possibility that some parcels have higher SF correspondence due to high SC or FC degree, i.e., having more WM connections or greater functional similarly with more parcels, respectively (Fig. S5). We determined there was an insignificant relationship with SC-Count degree, suggesting the number of WM streamlines per parcel had minimal effect on SF correspondence. However, we found there was a significant relationship with FC-Neurosynth degree, suggesting parcels that were functionally similar

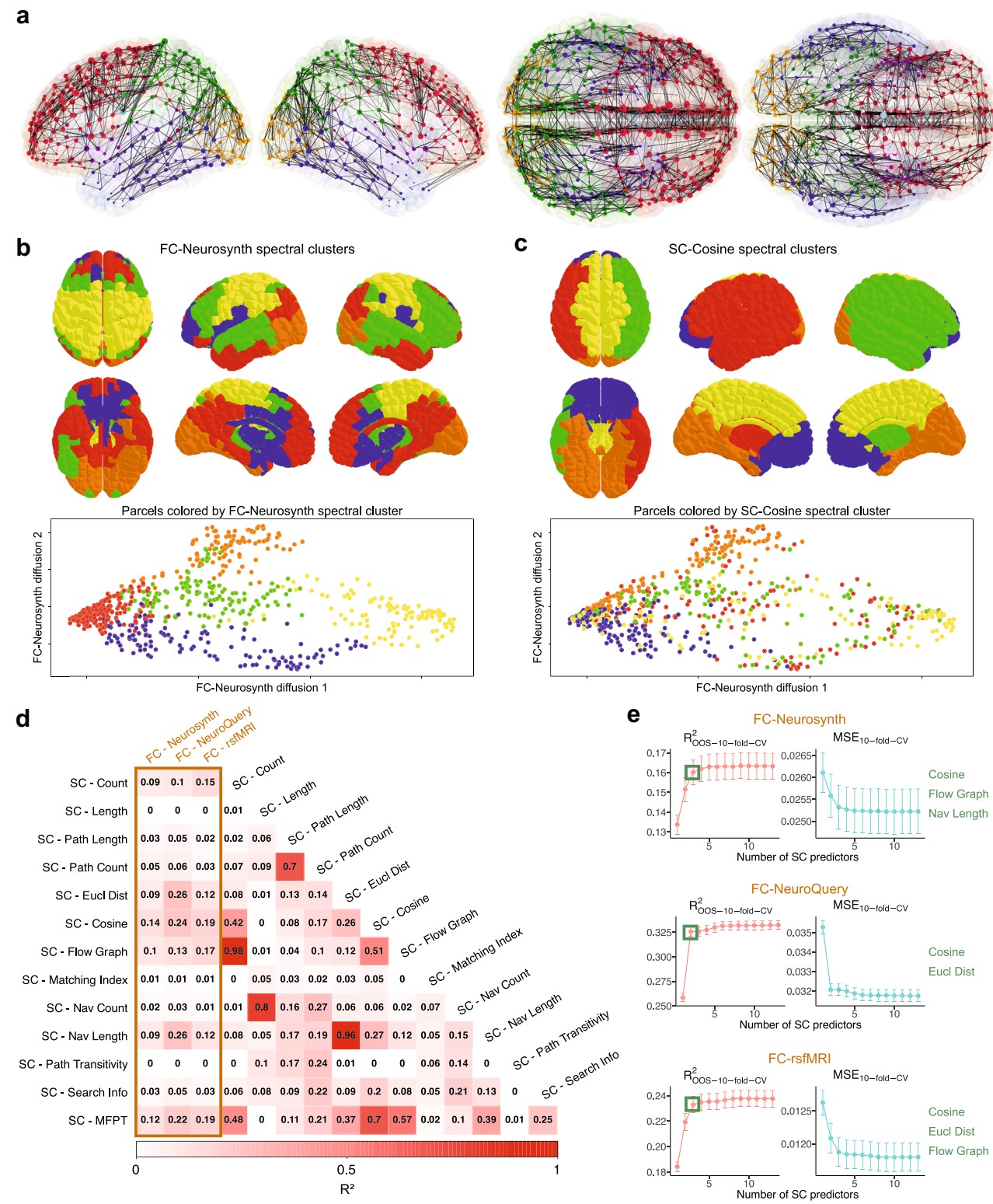

to many other parcels in Neurosynth generally had lower SF correspondence. This was not as significantly the case for FC-rsfMRI degree or FC-NeuroQuery degree, potentially due to their activation data being less sparse than that of Neurosynth.

## Structure-function correspondence predictably varies by specific function

Recognizing the principal advantage of using meta-analytic FC across many isolable functional terms, we next analyzed how SF

correspondence varies by specific function. For this effort, we chose FC-Neurosynth over FC-NeuroQuery due to our intended use of word embeddings, which compromise statistical independence with the latter given NeuroQuery's word embedding-based processing of activation data. To compute SF correspondence for a specific functional term, the active (i.e., non-zero activation in Neurosynth) and inactive parcels were determined. We then executed a Wilcoxon rank-signed test to compare a vector of WM streamline counts (i.e., SC-Count) between active-active parcel pairs and another vector of WM

**Fig. 2 | Global comparison between structural and functional connectivities.**
**a** Network graph of the brain (left to right: left lateral, right lateral, superior, basal) with nodes representing all 696 parcels and edges representing the binarized end-to-end WM connections (only top 5000 shown for clarity). Node size corresponds to structural connectivity (SC)-Count degree. Node color reflects brain region. Edge weight corresponds to functional connectivity (FC)-Neurosynth values. **b** Spectral clustering of parcels based on FC-Neurosynth (top). Two-dimensional diffusion map for FC-Neurosynth with each parcel colored according to FC-Neurosynth spectral cluster designation (bottom). **c** Spectral clustering of parcels based on SC-Cosine (top). Two-dimensional diffusion map for FC-Neurosynth with each parcel colored according to SC-Cosine spectral cluster designation (bottom). **d** $R^2$ values

from linear regression models evaluating the relationship between specific SC and FC values across the whole brain. All estimates within the gold box represent structure-function relationships. **e** Mean out-of-sample adjusted $R^2$ values across 10 cross-validation folds from linear regression models to predict either FC-Neurosynth, FC-NeuroQuery, or FC-rsfMRI with varying number of SC predictors. Comparable plots for mean squared error (MSE) are also shown. For each cross-validation fold and for each number of SC predictors, the optimal model was selected via exhaustive search. The list of terms in green reflect the optimal SC predictors that maximize adjusted $R^2$, beyond which adding more SC predictors does not significantly improve performance for that FC type. Points denote mean values and error bars denote +/−SEM. Source data are provided as a Source data file.

streamline counts between active-inactive parcel pairs for each functional term (Fig. 4a). Plotting each term by its resulting *p* value and fold change dimensionalized SF correspondence (Fig. 4b; see "Methods"). Applying significance thresholds, functional terms with high SF and low SF were selected; the labeled terms are more extensively shown in Fig. S6. Among those greater than the fold change threshold, the functional terms with most significant *p* values were "visual", "semantic", and "sensorimotor". Among those with significant *p* values, the functional terms with the greatest fold change were "sensorimotor", "foot", and "vision"; those with the least were "sensitivity", "retention", and "spectrum". Altogether, we observed functions related to sensory-motor, perceptual, and language tasks have higher SF correspondence.

We next leveraged NLP to assess if functional terms with higher SF correspondence clustered together when mapped in semantic space, derived from computing word embeddings of the terms based on a corpus of abstracts from Neurosynth (Fig. 4c). The first components of both tSNE and PCA in dimensionally-reduced semantic space significantly discriminated between high SF and low SF terms (Fig. 4d), suggesting the semantics of functional terms fundamentally organize their SF correspondence in the brain.

Next, we more precisely evaluated how the semantics differed between functional terms with high SF and those with low SF. Considering we observed higher SF correspondence localized to sensory and motor cortex in Fig. 3, we were motivated to reference a sensory-motor ratings resource from Brysbaert et al. to assign a so-called concreteness score to each functional term[22]. Concreteness scores were given in this prior study by asking scorers to rate words based on involvement of senses and motor responses. We found terms with high SF on average had significantly higher concreteness (i.e., sensory-motor) scores (Fig. 4e), reinforcing the spatial findings displayed in Fig. 3. Moreover, we also evaluated how using SC-Cosine rather than SC-Count affects this analysis, revealing a largely similar grouping of high SF and low SF terms separable by fundamental embeddings (Fig. S7a, b).

As opposed to discretizing words as either high SF or low SF, we also computed how these predictors relate to the fold change value as a continuous measure of SF (Fig. S8a–d); the same relationships shown in Fig. 4b–e were reinforced. Moreover, the fold change value of SF did not strongly relate to the mean distance between active parcels for a functional term (Fig. S8e), suggesting the proximity of activation areas was not superficially driving SF correspondence.

To assess how well semantics of functional terms in natural language can actually *predict* their SF correspondence, word embeddings of the functional terms were used as inputs to either a support vector machine (SVM) binary classifier or a fully connected neural network for regression (Fig. 4f). The SVM classified functional terms as either high SF or low SF. The fully connected feedforward neural network predicted the SF fold change of the functional terms. The SVM performed well with a mean AUC of 0.85. Moreover, the neural network performed modestly with a mean out-of-sample $R^2$ value of 0.15 and a Spearman *r* of 0.44. Altogether, both machine learning

implementations demonstrate how semantic embeddings can be used to predict the SF relationship elicited by specific functions.

## Structure-function correspondence unfolds along macroscale functional gradients

In addition to analyzing individual functional terms, we next explored macroscale functional gradients that correlate with SF correspondence across the brain. Using both dimensionality reduction and additional NLP techniques, we analyzed how parcel-wise metrics correlate with the SF correspondence pattern derived from FC-Neurosynth shown in Fig. 3a.

A prior study identified the first dimension of the diffusion map for FC-rsfMRI to correlate well with SF correspondence by region[10]. We replicated this key finding for FC-Neurosynth, computing relatively strong correlation between SF correspondence and the first component of the diffusion map for FC-Neurosynth ($R^2 = 0.2$, Spearman $r = 0.44$) (Fig. 5a). We explored another dimensionality reduction technique, PCA, and found that the first principal component derived from Neurosynth functional activation (FA) data also significantly correlated with SF correspondence ($R^2 = 0.15$, Spearman $r = 0.35$) (Fig. 5b), although with less explanatory power than the first diffusion component. The prior study hypothesized that a unimodal-to-transmodal gradient was the central driver of the FC diffusion component, which in turn correlates well with the SF correspondence gradient[10]. However, this modality hypothesis has not been directly testable with rsfMRI-derived FC. Thus, we exploited the diversity of our meta-analytic FC along many isolable functional dimensions. Moreover, we used NLP techniques based on semantics of the functional terms to identify additional macroscale functional gradients that predict SF correspondence across the brain.

To operationalize the previously hypothesized unimodal-to-transmodal gradient[10], we defined unimodality as low functional diversity and transmodality as high functional diversity. To quantify functional diversity, we began by determining the word embeddings of the ten functional terms with the highest Neurosynth activation values for each parcel. We did not consider all activated terms for each parcel to reduce noise from Neurosynth. Pairwise distances between word embeddings capture semantic differences[23]. Thus, we computed functional diversity for each parcel as the weighted mean distance between word embeddings of the ten most activated terms and their corresponding centroid. We discovered functional diversity (i.e., uni-/transmodality) negatively correlates with SF correspondence; however, the relationship was weak ($R^2 = 0.02$, Spearman $r = -0.13$) (Fig. 5c).

Next, we explored how the extent of sensory-motor function in each parcel correlated with SF correspondence. As in Fig. 4e, we referenced the concreteness ratings resource from Brysbaert et al. to assign concreteness (i.e., sensory-motor function) scores for each parcel[22]. We found the extent of sensory-motor function by parcel positively correlates with SF correspondence ($R^2 = 0.23$, Spearman $r = 0.44$) (Fig. 5d), with even higher explanatory power as the first diffusion component.

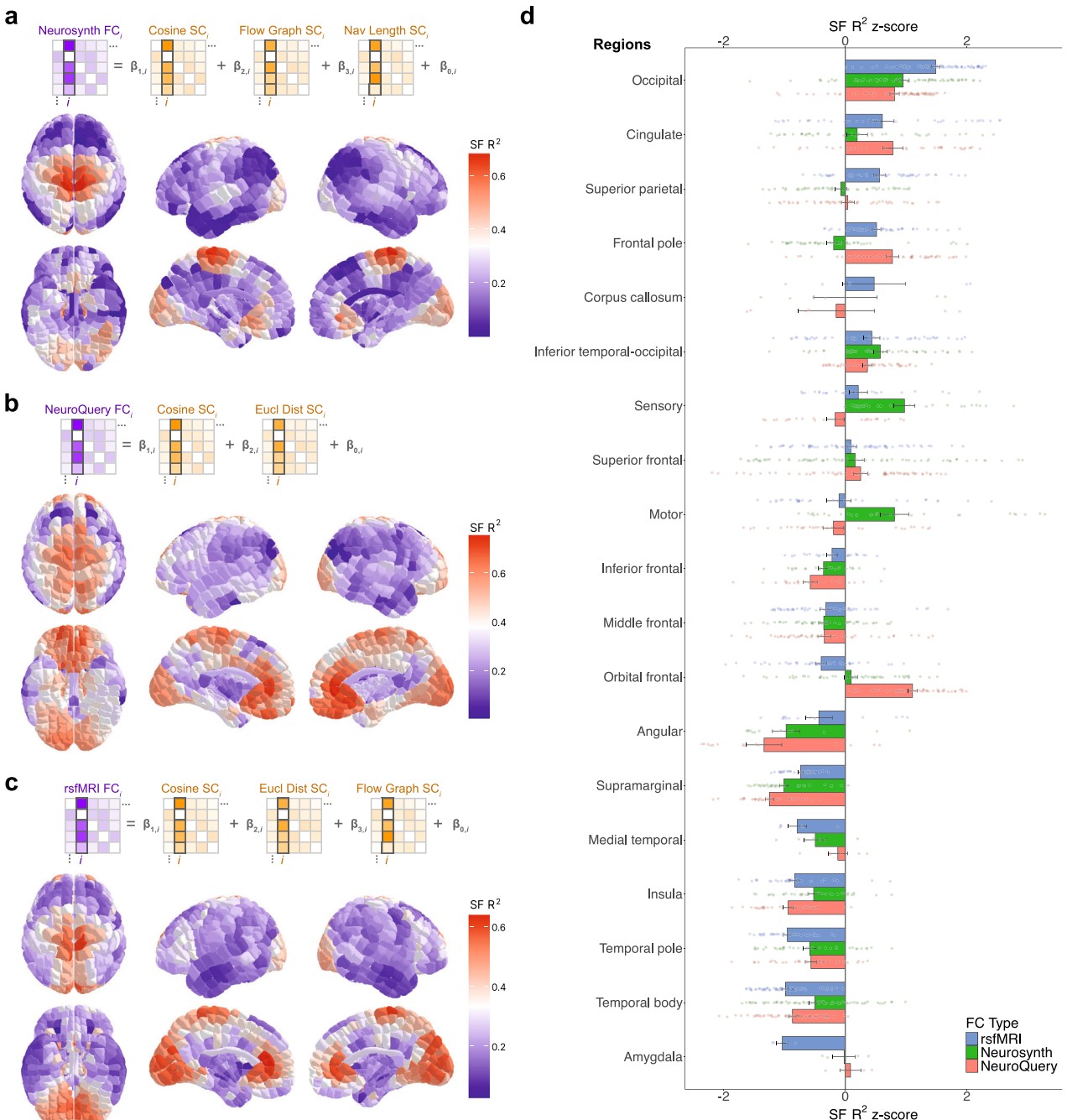

**Fig. 3 | Structure-function correspondence varies by localization and differs depending on FC type. a** Matrix schematic demonstrating how strength of structure-function (SF) correspondence in each parcel was computed as the adjusted $R^2$ value resulting from a multiple linear regression model relating functional connectivity (FC)-Neurosynth with the optimal structural connectivity (SC) measures selected in Fig. 2e (top). Pattern of SF correspondence across the brain, with parcels colored by these $R^2$ values (bottom). **b** Analogous to Fig. 3a for FC-

NeuroQuery. **c** Analogous to Fig. 3a for FC-rsfMRI. **d** Bar chart of SF $R^2$ z-scores by region for each FC type, with each bar representing the mean +/− SEM among constituent parcels for a particular region ($N = 19$). Each point (total $N = 696$) reflects a constituent parcel for a region. z-scores were used to enable more effective comparisons between FC types. Source data are provided as a Source data file.

Lastly, we identified three more functional gradients that correlated with SF correspondence: the extent of perceptual, cognitive, and biological functions. We referenced the psychometric categorization of words found in the Linguistic Inquiry and Word Count (LIWC2015) dictionary[24]. Note the biological category in LIWC2015 denotes words related to core body, health, sexual, and ingestion processes. We found the extent of perceptual function by parcel positively correlates with SF correspondence ($R^2 = 0.07$, Spearman

$r = 0.34$) (Fig. 5e), the extent of cognitive function by parcel negatively correlates with SF correspondence ($R^2 = 0.11$, Spearman $r = -0.33$) (Fig. 5f), and the extent of biological function by parcel positively correlates with SF correspondence ($R^2 = 0.13$, Spearman $r = 0.35$) (Fig. 5g). With respect to the multicollinearity of these predictor variables, we found they have variance inflation factors (VIFs) ranging from 1.1 to 3, suggesting relatively limited multicollinearity.

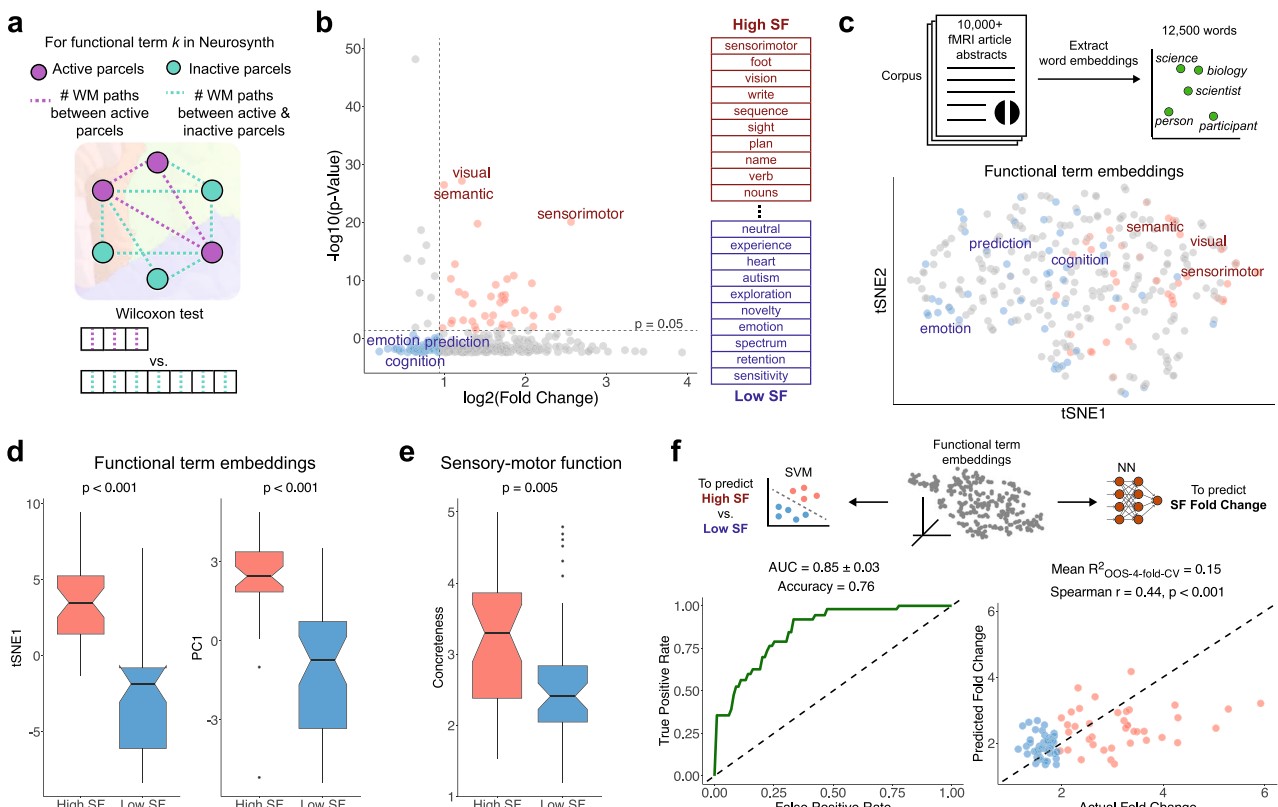

**Fig. 4 | Structure-function correspondence predictably varies by specific function. a** Schematic illustrating how structure-function (SF) correspondence was evaluated for each specific functional term. For each of the 334 functional terms in Neurosynth, a Wilcoxon rank-signed test compared a vector of SC-Count values (i.e., number of white matter (WM) paths) between active-active parcel pairs and another vector of SC-Count values between active-inactive parcel pairs for each term. **b** The resulting $p$ values and fold changes are plotted, with significance thresholds at $p = 0.05$ and $\log_2$(Fold Change) $\approx 0.94$, reflecting the $20^{th}$ percentile cutoff among significant terms. A select number of functional terms are labeled in the plot, shaded red to indicate high SF or blue to indicate low SF. Table shows opposite ends of functional terms list ranked by fold change. **c** Word embeddings were computed via word2vec from a corpus of 12,909 web-scraped fMRI article abstracts used in Neurosynth. Schematic with mock word embeddings illustrates the basic idea underlying embedding space: words closer to each other in this space are more semantically similar in natural language. Plot below shows tSNE visualization of word embeddings of the 334 functional terms, colored by the same convention in Fig. 4b. **d** Boxplots demonstrating significant fundamental differences in embedding space between high SF functional terms ($N = 39$) and low SF functional terms ($N = 49$); the first component from tSNE shown left, and the first component from PCA shown right. Two-sided Wilcoxon signed-rank tests were performed; $p$ values were Bonferroni-corrected (exact $p$ values are $5.84 \times 10^{-13}$ (left),

$4.23 \times 10^{-11}$ (right)). **e** Boxplot demonstrating significant difference in concreteness score between high SF functional terms ($N = 34$) and low SF functional terms ($N = 46$). Concreteness scores for each term were sourced from Brysbaert et al. and reflect the degree of sensory and motor responses involved[22]; 8 terms were not featured in this source and were thus omitted from analysis. Two-sided Wilcoxon signed-rank test was performed. All boxplots have a box that signifies the interquartile range (IQR; 25th percentile to 75th percentile), a center bar that denotes the median, whiskers that extend up to 1.5 × IQR, and a notch that extends 1.58 × IQR/√n, where $n$ is the sample size for that condition, to estimate the 95% confidence interval. **f** Schematic demonstrating the two ways how word embeddings of the functional terms were used as inputs to either (1) a support vector machine (SVM) to classify functional terms as high SF or low SF, or (2) a fully connected feedforward neural network (NN) to predict SF fold change. The mean ROC curve over four cross-validation folds for the SVM is shown left with its AUC and mean accuracy reported; note that high SF is the positive class. The predicted vs. actual value plot over four cross-validation folds for the neural network is shown right with the mean out-of-sample $R^2$ value, Spearman $r$ value, and $p$ value (exact $p$ value of $1.93 \times 10^{-5}$) from the correlation test reported. Both SVM and neural network results demonstrate how word embeddings can be used to predict the SF relationship elicited by specific functions in the brain. Source data are provided as a Source data file.

## Organization of cortical thickness relates to structure-function correspondence

Prior research indicated structural measures like intracortical myelination[25] and laminar differentiation[26] correlate with functional organization in the cortex. Moreover, recent evidence has suggested cortical thickness follows similar organizational axes as myeloarchitecture, cytoarchitecture, and functional networks[27–29]. Here, hypothesizing the importance of neuroanatomical features to SF coupling, we assessed the correlations of cortical thickness with SF correspondence, as well with macroscale functional gradients. We first computed an average cortical thickness map in Yale Brain Atlas space for 200 healthy young adults from HCP (Fig. 6a). We omitted the six corpus callosum parcels from this analysis, thus, bringing the total Yale Brain Atlas parcels evaluated to 690.

In light of recent evidence suggesting cortical thickness follows key gradients that run coronally and axially[27], we found cortical thickness correlates with coronal ($R^2 = 0.01$, Spearman $r = 0.14$) and, especially, axial ($R^2 = 0.44$, Spearman $r = -0.67$) Cartesian coordinates of parcels. We observed a negative, albeit weak, correlation between cortical thickness and SF correspondence ($R^2 = 0.03$, Spearman $r = -0.2$) (Fig. 6b). More localized comparisons revealed that the strongest correlation between cortical thickness and SF correspondence exists in the parietal lobe ($R^2 = 0.61$, Spearman $r = -0.81$) (Fig. 6c).

Comparing with the macroscale functional gradients in Fig. 5, cortical thickness significantly correlates with the first component of the FC diffusion map ($R^2 = 0.23$, Spearman $r = -0.46$) (Fig. 6d), functional diversity ($R^2 = 0.04$, Spearman $r = 0.22$) (Fig. 6e), sensory-motor

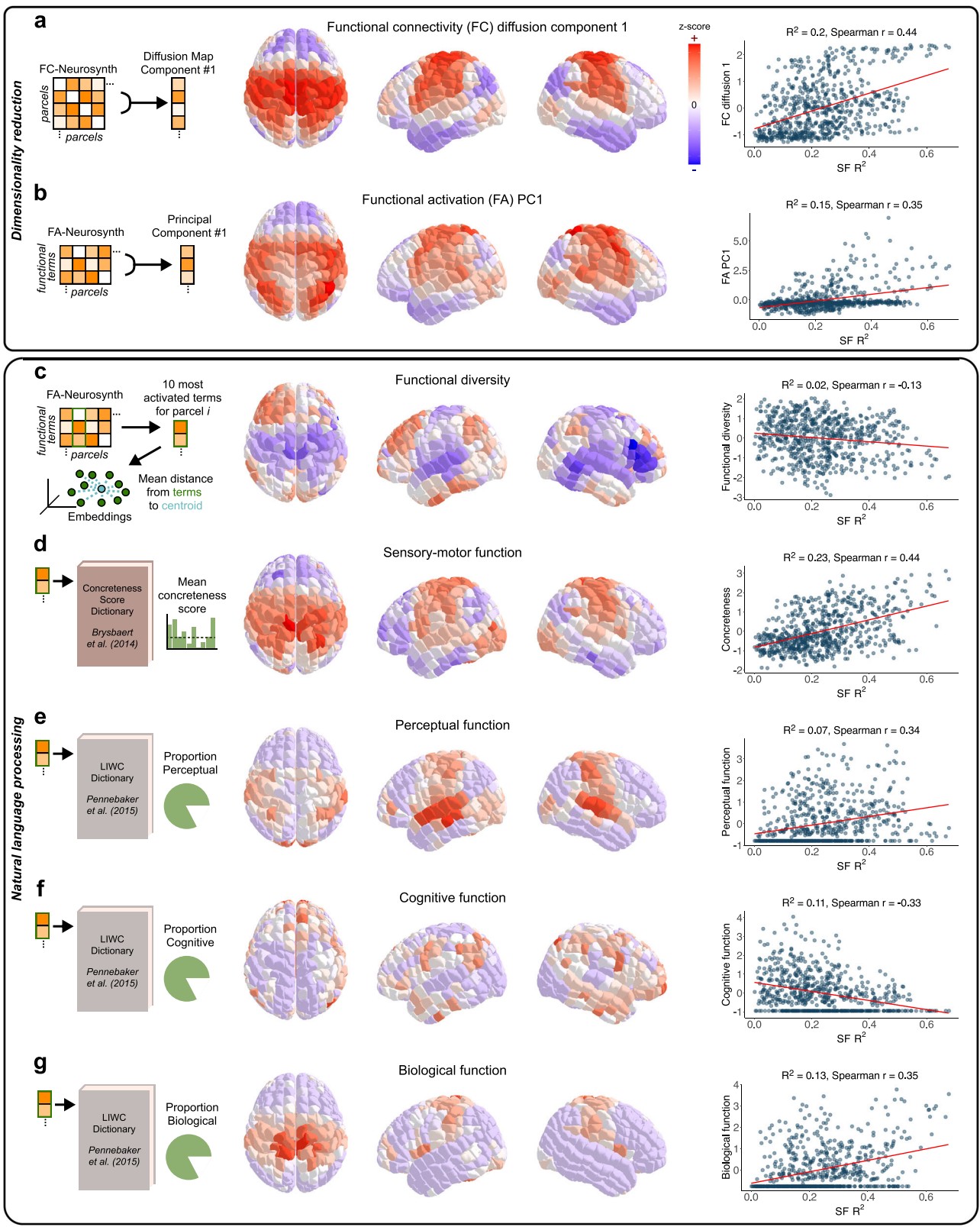

function ($R^2 = 0.12$, Spearman $r = −0.33$) (Fig. 6f), and perceptual function ($R^2 = 0.03$, Spearman $r = −0.21$) (Fig. 6g). Cortical thickness demonstrates insignificant correlations with the two other functional gradients derived from psychometric categorization, i.e., cognitive function and biological function. These correlations that are significant suggest some degree of multicollinearity between cortical thickness and other correlates of SF correspondence. Despite having relatively low VIFs ranging from 1.2 to 3.4, multicollinearity concerns prevent definitive conclusions about the way in which cortical thickness directly influences SF correspondence; nevertheless, our study reports an association with cortical thickness pertinent to SF coupling and other macroscale functional gradients.

**Fig. 5 | Macroscale functional gradients correlate with structure-function correspondence across the brain. a** Schematic demonstrating the first component of the diffusion map for FC-Neurosynth (left). Pattern of this first diffusion component across the brain is mapped, with parcel color representing z-score (center). Correlation between the first component of the FC-Neurosynth diffusion map and the parcel-wise SF $R^2$ values as shown in Fig. 3A; $R^2$ and Spearman $r$ values noted (right). **b** Schematic demonstrating the first principal component for Neurosynth functional activation (FA-Neurosynth) data as introduced in Fig. 1e (left). Pattern map across the brain (center) and correlation with SF $R^2$ (right) are shown for the first FA principal component (PC) analogous to Fig. 5a. Note that parcel color here denotes log-transformed z-score value; log was applied to reduce skewness. **c** Measures derived with NLP techniques selected the ten functional terms with the highest Neurosynth activation values for each parcel. Functional

diversity was quantified as the weighted mean distance between word embeddings of the ten most activated terms and their corresponding centroid for each parcel (left). Pattern map across the brain (center) and correlation with SF $R^2$ (right) are shown for functional diversity analogous to Fig. 5a. **d** Extent of sensory-motor function was quantified as the weighted mean concreteness score for the ten most activated terms for each parcel. Concreteness scores were sourced from Brysbaert et al.[22] (left). Pattern map across the brain (center) and correlation with SF $R^2$ (right) are shown for concreteness scores analogous to Fig. 5a. **e** Extent of perceptual function was quantified as the weighted proportion of the ten most activated terms labeled as "perceptual" in LIWC for each parcel[24] (left). Pattern map across the brain (center) and correlation with SF $R^2$ (right) are shown for perceptual function scores analogous to Fig. 5a. **f** Analogous to Fig. 5e for cognitive function. **g** Analogous to Fig. 5e for biological function. Source data are provided as a Source data file.

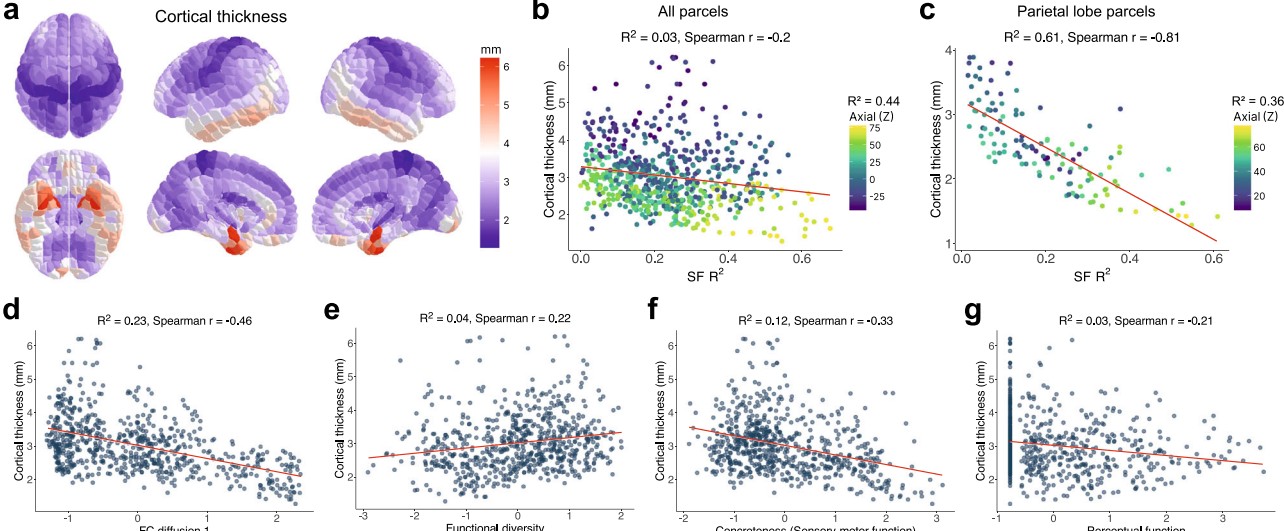

**Fig. 6 | Cortical thickness correlates with structure-function correspondence and macroscale functional gradients. a** Map of cortical thickness values for each parcel. **b** Significant negative correlation between cortical thickness and SF $R^2$ values as shown in Fig. 3a; $R^2$ and Spearman $r$ values noted above plot. Parcels colored according to axial Cartesian coordinates, and its $R^2$ value with cortical thickness noted above legend. **c** The strongest correlation between cortical

thickness and SF $R^2$ values was found in the parietal lobe, similarly colored according to axial coordinates. Correlations between cortical thickness and four macroscale functional gradients introduced in Fig. 5: **d** the first component of the FC-Neurosynth diffusion map, **e** functional diversity, **f** sensory-motor function, and **g** perceptual function. $R^2$ and Spearman $r$ values reported. Source data are provided as a Source data file.

## Discussion

Despite prior research on the spatial gradient of SF correspondence in the human brain[10,14,26,30], the effect of different specific functions on the SF relationship has remained unknown. Our analysis of large data repositories reveals converging evidence that SF correspondence unfolds along macroscale functional and neuroanatomical gradients that follow a sensory-fugal organizational axis, with higher correspondence in primary sensory and motor cortex for perceptual and motor functions, and lower correspondence in association cortex for cognitive functions.

To compute SF correspondence across more than 300 specific functions, we sourced functional data from the meta-analytic repositories Neurosynth[18] and NeuroQuery[19] and transformed them into high-resolution atlas space. The former was included because it has been an essential meta-analytic resource for neuroscience research for more than a decade, the latter as a secondary point of comparison because it offers an updated repository with smoothed data. Although meta-analytic functional data presents many limitations, namely, minimally interpretable topic analysis and lack of individual-specific data, we nevertheless recognized its value for diverse functional activation data. Nevertheless, considering other studies have observed individual differences in SF coupling[31], future research should explore

individual-specific analyses based on alternative functional databases. The observed similarity in the SF correspondence gradients derived from meta-analytic FC (i.e., FC-Neurosynth or FC-NeuroQuery) and FC-rsfMRI suggests the former, despite being sourced from noisy meta-analysis, aligns moderately well with the traditional approach of analyzing FC patterns of a small resting-state cohort. Meta-analytic FC data, importantly, has the unique benefit of spanning many isolable functional dimensions, allowing for analysis by specific function. Coupled with WM tractograms for 1065 subjects from HCP dMRI data, our results demonstrate how SF correspondence varies across specific regions and functions in the human brain.

This study contributes to a growing body of evidence that our understanding of SF correspondence is incomplete. Many early reports analyzed the relationship between structure and function at the whole-brain level[11,32]. Like these studies which used FC-rsfMRI measures, the findings reported here based on meta-analytic FC indicate an imperfect SF correspondence globally, with the majority of functional variance not explained by direct structural metrics[11], motivating further investigation into new analytical methods and experimental techniques besides dMRI for SC and fMRI for FC.

We demonstrate SF correspondence is not spatially uniform across the brain. We performed multilinear regression model selection

via exhaustive search over ten cross-validation folds to select the optimal SC measures to predict FC and in turn quantify SF correspondence at every parcel of a high-resolution anatomical atlas[16]. Our results largely agree with a prior study which used FC-rsfMRI, illustrating higher SF correspondence in primary sensory and motor cortex especially along the longitudinal fissure, and lower in association cortex. However, using FC-Neurosynth importantly yielded an even more pronounced discrepancy between these regions. Using FC-NeuroQuery led to a marked increase in SF correspondence in most regions, most notably in the orbitofrontal cortex. We hypothesize the overall higher SF correspondence when using FC-NeuroQuery is a consequence of its use of semantic smoothing to generate activation data, which in turn likely enhances the predictive power of Euclidean distance in our analyses. The most notable difference between meta-analytic FC and FC-rsfMRI was observed in the amygdala, a region involved in emotional learning and behavior including fear and anxiety[33]. Like other regions with relatively high SF correspondence, the activity of the amygdala is evolutionarily important for efficient processing of environmental stimuli[34]. Prior SF studies using rsfMRI did not emphasize the amygdala, potentially due to low activation during resting state. Overall, this demonstrates how incorporating specific functions in SF analysis can highlight more biologically and evolutionarily meaningful patterns.

We next leveraged NLP techniques to discover functional terms with different SF correspondence strengths have fundamentally different semantics. We found that semantic embeddings of functional terms could successfully predict their SF correspondence strength. This suggests humans' conception about the extent of difference between meanings of functional words mirrors the extent of difference in the SF patterns they elicit in the brain. Hence, this study reveals a notable convergence of the neural basis of functions and our conscious discernment of these functions as expressed in natural language: we potentially conceive functional terms as different due to the different network properties they elicit in our brain; alternatively or in combination, conceived diversity of functional terms could only manifest as a consequence of variable SF correspondence. To further define what is driving the semantic axis discriminating SF correspondence, we demonstrated functions with greater sensory-motor characteristics have higher SF correspondence on average.

In addition to analyzing individual functional terms, we also explored macroscale functional gradients that predict SF correspondence across many parcels in the brain. Using dimensionality reduction, we found both the diffusion map component from FC-Neurosynth and principal component from FA-Neurosynth (i.e., functional activation data of Neurosynth) significantly discern SF correspondence across parcels. Both the diffusion map component and principal component, by mathematical definition[35,36], represent an underlying representative dimension of brain function; thus, we next analyzed the contributions of specific functional characteristics that were potentially driving these global gradients. Using NLP approaches, we discovered higher SF correspondence in regions that have less functional diversity (i.e., unimodality), more sensory-motor function, less higher-order cognitive function, more perceptual function, and more basic biological function. Overall, SF correspondence is greater for externally-oriented sensory-motor functions, and lower for internally-oriented cognitive functions.

Altogether, the localized SF correspondence gradient from primary sensory to association areas aligns with a so-called sensory-fugal gradient[37], shown previously to serve as the organizational axis of intracortical myelination, laminar differentiation, and cortical thickness[26,27,38]. Our findings reinforce this axis for cortical thickness organization, as we observed cortical thickness associated mildly with our SF correspondence gradient, with thinner cortex having higher SF correspondence on average. We also observed cortical thickness to be associated with select macroscale functional gradients, including the

FC diffusion map component, functional diversity, sensory-motor function, and perceptual function. One hypothesis is thinner cortex prioritizes fewer, more functionally-essential direct WM connections; however, with cortical thickness potentially being colinear with other predictors of SF correspondence, future research is warranted to further explore how neuroanatomical features drive SF coupling.

Why do externally-oriented, perceptual functions in primary sensory and motor cortex have greater SF coupling? One possibility could once again involve the evolutionary cortical expansion of the human brain along the sensory-fugal axis[12]. As mammalian brains evolved and evolutionary pressures shifted from lower-level environmental processing to higher-order cognitive functions[12,39,40], it is possible that useful functionality achieved from direct structural connections had been exhausted, and organisms established more indirect structural connections to enable emergent functional properties in what would become the association cortex. A second possibility could pertain to the dual processing theory of automatic and controlled processing[41]. More direct WM connections potentially evolved to enable faster responses during lower-level functions especially important for survival, and a multitude of indirect WM connections evolved to optimize response time for higher-level functions[42,43]. This explanation is complemented by the finding that sensory-motor regions have lower dynamic range, enabling them to quickly transition between low and high activity levels[44].

Our results altogether explore how the relationship between structure and function varies by specific region and function in the human brain. Future research using new data modalities, individualized connectomes, additional cytoarchitectural features, and specialized connectivity metrics could be essential for more comprehensively linking structure to function. Nevertheless, by identifying the sensory-fugal dimension as a central organizational axis of SF correspondence across many functions, this study advances our understanding of the diversity and evolution of network properties in the human brain.

## Methods
### Atlas
All analyses used the Yale Brain Atlas (YBA). YBA is a high-resolution anatomical parcellation of the human brain. YBA consists of parcels spanning the cortex, hippocampus, and amygdala. It also includes a comprehensive anatomical nomenclature to facilitate communication and aid analyses[16]. For all analyses except those involving cortical thickness, the 696-parcel-version of YBA was used, which includes 6 parcels in the corpus callosum; these parcels were excluded from cortical thickness analysis. Atlas position and indices files for both the 690- and 696-parcel-version of YBA used in this study can be downloaded from https://github.com/evancollins1/brain_structure_function/tree/master/data. YBA can be interactively viewed at https://yalebrainatlas.github.io/YaleBrainAtlas/.

### Structural connectivity
Structural data was acquired from Yeh[45], which includes processed tractograms derived from dMRI data for 1,065 young adult subjects (575 females, 490 males; mean age 28.74 years; age range from 22 to 37 years). The dMRI data originated from HCP and was converted to the DSI Studio file format. Yeh reconstructed the dMRI data in the MNI common space using q-space diffeomorphic reconstruction[45,46]. For our analysis, a deterministic fiber tracking algorithm was used, and a seeding region was placed across the whole brain in DSI Studio. The anisotropy threshold was 0.05208 (0.6 × Otsu's threshold). The angular threshold was 60 degrees. The step size was 0.5 voxels. Tracks with length shorter than 10 mm or longer than 300 mm were discarded. A total of 1,000,000 streamlines were generated for each subject, and two iterations of topology-informed pruning were performed to remove false positive streamlines[47]. The SC matrix for WM

streamline count was calculated for each subject by counting streamlines with endpoints in distinct parcel pairs. To compute a representative group-level SC matrix for WM streamline count (i.e., SC-Count), we averaged subject-specific WM streamline matrices. As was done in a prior study[9], we normalized SC-Count by dividing the averaged streamline count between any two parcels by the geometric mean volume of the two parcels. We experimented with differing zeroing thresholds (i.e., there must exist non-zero WM streamlines in more than some $X\%$ of subjects for inclusion in the group-level matrix) and found maximal SF correspondence without any zeroing threshold (Fig. S2A). In addition, we evaluated the effect of weighting SC-Count either by multiplying or dividing by WM length and found maximal SF correspondence without any weighting (Fig. S2B). Thus, with our primary interest being SF correspondence, we maintained the unthresholded, unweighted, volume-normalized SC-Count group-level matrix. Nevertheless, further research is encouraged to explore how thresholding and weighting better capture underlying group-level neuroanatomy.

Next, the SC matrix for WM length between structurally connected parcel pairs was computed for each subject by taking the arithmetic mean of streamline lengths between them. The 1065 WM length matrices were then averaged to yield a single group-representative matrix (i.e., SC-Length). Lastly, the SC matrix for Euclidean distance (i.e., SC-Eucl Dist) was calculated as the Euclidean distance between centroids of parcels. These three group-level SC matrices—SC-Count, SC-Length, SC-Eucl Dist—were next inputted into network analysis scripts from Zamani Esfahlani et al.[9] to yield an additional eight group-level SC matrices. These additional weighted SC metrics are cosine similarity (i.e., SC-Cosine), flow graphs (i.e., SC-Flow Graph), matching index (i.e., SC-Matching Index), navigation count and length (i.e., SC-Nav Count, SC-Nav Length), path transitivity (i.e., SC-Path Transitivity), search information (i.e., SC-Search Info), and mean first passage time (i.e., SC-MFPT). Mathematical definitions of each can be found in Zamani Esfahlani et al.[9].

In addition to these SC metrics, we computed two additional SC metrics: path length (i.e., SC-Path Length) and path count (i.e., SC-Path Count). Path length is defined as the total WM length of the shortest path connecting two parcels. Path count is the minimum number of end-to-end streamline connections needed to connect two parcels.

We strove to include these 13 different SC measures in order to comprehensively capture network information corresponding to neuroanatomical organization. Although geodesic distance is commonly included as an SC measure[13], our use of a volumetric atlas with parcels in deep structures precluded us from using this surface-based measure[48]. Moreover, because we sought to conduct analyses at the whole-brain level rather than at the level of single hemispheres, we included corpus callosum parcels to better capture interhemispheric connections.

## Neurosynth-derived functional connectivity
Functional data for Neurosynth was acquired from the GitHub repository at https://github.com/neurosynth/neurosynth-data. Neurosynth has collated fMRI activation data from 14,371 peer-reviewed articles and features voxelwise z-score maps of activations across 1334 terms[18]. The z-scores by term in Neurosynth were calculated from a one-way ANOVA assessing whether the proportion of studies that tabulate activation at a certain voxel differs from the proportion that would be anticipated if activations were uniformly distributed in gray matter. The 1334 terms emerge from topic analysis of the web-scraped articles, which include both rsfMRI- and diverse task-based studies. The number of terms to keep in Neurosynth (i.e., 1334) was determined by a minimum threshold for the number of studies involving each term. For our analyses, a team of expert neuroscientists further reduced this list to 334 functional terms by removing terms that described locations (e.g., "prefrontal") and averaging terms that shared roots (e.g., "emotional" and "emotion"). The limitation of a topic-analysis approach for

defining functions is mainly one of specificity; however, prior analysis suggests these NLP-derived functional terms are sufficiently representative proxies of functional processes[18]. For each of the 334 functional terms, we averaged activation z-scores from the voxels contained within each YBA parcel. The resulting data included the averaged z-scores associated with 334 functional terms for 696 atlas parcels; we call this the Neurosynth functional activation (FA-Neurosynth) data. To produce connectivity data with dimensions 696 by 696, cosine similarity was calculated between the functional vectors of each pair of parcels; we call this the FC-Neurosynth data. We chose the cosine similarity metric to minimize the effect of magnitude differences across parcels; cosine similarity importantly captures only the relative differences in activation across terms for each parcel. We considered thresholding FC to match the sparsity of SC; however, this would have resulted in a significant loss of data. We instead opted to report correlations using rank-based approaches.

## NeuroQuery-derived functional connectivity
Functional data for NeuroQuery was acquired using the model included in the *neuroquery* Python package. Much like Neurosynth, NeuroQuery is a meta-analytic tool for fMRI activation data across a variety of terms. NeuroQuery expands Neurosynth namely through a larger, updated dataset of text (i.e., about 75 million words) and the use of semantic smoothing based on word embeddings to synthesize activation maps[19]. We evaluated the activation data of NeuroQuery using the same 334 functional terms that we subsetted from Neurosynth. As with Neurosynth, for each of the 334 functional terms, we averaged NeuroQuery activation z-scores from the voxels contained within each YBA parcel. The resulting data included the averaged z-scores associated with 334 functional terms for 696 atlas parcels, forming the NeuroQuery functional activation (FA-NeuroQuery) data. Cosine similarity was used to compute connectivity values (i.e., FC-NeuroQuery). For our downstream analyses involving word embeddings, we chose to reference FC-Neurosynth data rather than FC-NeuroQuery to not compromise statistical independence considering NeuroQuery uses word embeddings to generate and process activation data.

## rsfMRI-derived functional connectivity
For rsfMRI data acquisition, imaging was performed on a 3-Tesla Siemens Trio scanner (Siemens Medical Systems, Erlangen, Germany) using a 64-channel head coil for 34 subjects (17 females, 17 males; mean age 33 years; age range 18 to 55 years) (IRB 1003006485 & 0702002395). Informed consent was signed by all subjects. Sex was based on that assigned at birth. Each subject was compensated for their participation. Each subject was positioned in the coil, and head movements were minimized with pillows. After a three-plane localizer, a high-resolution whole-brain T1-weighted three-dimensional magnetization-prepared rapid gradient echo volume scan was acquired for multisubject registration. An oblique axial T1 image parallel to the anterior commissure–posterior commissure (AC-PC) line was acquired. Two to three 5-min resting-state runs were then acquired. All scans were converted from Digital Imaging and Communication in Medicine format to NIfTI format. During the conversion process, six images were omitted to allow the signal to attain steady-state equilibrium between radio frequency pulsing and relaxation. Resting-state images were motion corrected using SPM8. Trials with linear motion that had a displacement more than 2 mm or rotation more than 3° were omitted. Artefact time courses related to cardiovascular and breathing effects (i.e., mean time courses from CSF and white matter) and global signal were regressed out. Bandpass filtering with a passband between 0.008 and 0.12 Hz was applied such that the output time course accurately represented the hemodynamic change of the BOLD signal. To take the individual subject data into a common reference space, we calculated sequential registrations within Yale BioImage Suite[49]: first, a linear registration between the individual subject's rsfMRI image and their anatomical image, and,

second, a nonlinear registration between the subject's anatomical image and the standard whole-brain template (MNI152 1 mm template). Mean rsfMRI activations across the entire time course were computed for each parcel. Finally, the connectivity matrix was obtained using the Fisher transformation of the Pearson correlation coefficient for each pair of parcels. Hence, the final output from this processing was a rsfMRI pairwise correlation connectivity matrix (i.e., FC-rsfMRI) of size 696 by 696 (parcels) for each of the 34 healthy subjects.

## Diffusion map and spectral clustering

Diffusion mapping is a nonlinear dimensionality reduction technique which computes a set of low-dimensional embeddings from high-dimensional data to map underlying manifolds[50]. Diffusion mapping has been used in prior studies investigating how connectivity patterns unravel in the brain[10,13]. In short, the diffusion map algorithm computes an embedding of the points based on the right eigenvectors of the powered Markov transition matrix $M^t$ where $M = D^{-1}W$ with $W$ as an affinity matrix and $D$ as the diagonal matrix of row sums[35]. To compute the diffusion map from our FC-Neurosynth data (computed from cosine similarities), we treated FC as a precomputed affinity matrix and set the diffusion parameter $t = 1$. The same was done for SC-Cosine for Fig. S3.

To perform spectral clustering of parcels based on the eigenvectors of the graph Laplacian for FC and SC, we used the Python library *scikit-learn*. We inputted the FC-Neurosynth data as a precomputed affinity matrix. We also considered applying a Gaussian kernel to construct the affinity matrix for FC; however, the spectral clusters produced were similar to those produced from the cosine affinity matrix, so we maintained the simpler cosine approach. The eigengap heuristic was used to determine an optimal number of FC-Neurosynth clusters, $k = 5$. The same procedure was applied to obtain spectral clustering results for FC-NeuroQuery and SC-Cosine. To allow for easy comparison between all FC and SC spectral clusters, $k = 5$ clusters were also used for FC-NeuroQuery and SC-Cosine.

We used spectral clustering and diffusion mapping of parcels together to visualize the different global patterns of SC and FC. FC spectral clusters (either FC-Neurosynth or FC-NeuroQuery) and SC spectral clusters exhibited limited alignment (Figs. 2b, c, S1a top). In addition, we used diffusion mapping of FC-Neurosynth (Figs. 2b, c, S1a bottom) and SC-Cosine (Fig. S3d–f) to visualize if lower dimensional manifolds could separate parcels by spectral cluster. Moreover, we found axial and coronal coordinates separate parcels in FC-Neurosynth diffusion space, suggesting a correlated continuum of function along these Cartesian dimensions, possibly reflecting a primary evolutionary axis in the brain (Fig. S3a–c). We observed SC-Cosine spectral clusters have limited organization in FC-Neurosynth diffusion map space (Fig. 2c bottom), once again suggesting minimal similarities in FC and SC organization globally. This lack of alignment is also the case when viewed in SC-Cosine diffusion map space (Fig. S3d–f). Altogether, this imperfect global correspondence between SC and FC motivated us to explore how SF correspondence varies by specific localization and function.

## Linear regression model selection

To assess the relationship between SC and FC, we first computed pairwise linear regression models between different SC and FC types across the whole brain. We report the pairwise in-sample $R^2$ values (Fig. 1d). Moreover, for each of these pairwise linear regression models, we computed mean out-of-sample $R^2$ values across ten cross-validation folds to assess predictive performance (Fig. S1c). Out-of-sample $R^2$ was calculated as

$$R^2_{OOS} = 1 - \frac{\sum_{i=1}^{N}(y_i - \hat{y}_i)^2}{\sum_{i=1}^{N}(y_i - \bar{y}_{OOS})^2}$$

for each cross-validation fold[51]. To generate the cross-validation folds, we implemented parcel-wise splitting, i.e., node-based splitting, to ensure the test set includes predictions for completely unseen relationships. Overall, we found minimal differences between the pairwise in-sample $R^2$ values and pairwise out-of-sample $R^2$ values.

Next, to select optimal multiple linear regression models to predict either FC-Neurosynth, FC-NeuroQuery, or FC-rsfMRI from some combination of SC measures, we computed mean out-of-sample adjusted $R^2$ values and mean squared error (MSE) values across ten cross-validation folds with varying number of SC predictors (Fig. 2e). This prediction-centric approach was chosen to prevent overfitting, which can occur when in-sample adjusted $R^2$ values are used for model selection (Fig. S1d). For each cross-validation fold and for each number of SC predictors, the model ultimately plotted was selected via exhaustive search to minimize the residual-sum-of-squares (RSS). For each FC type when considering across any number of SC predictors, we selected the optimal SC predictors that maximized out-of-sample adjusted $R^2$, beyond which adding additional SC predictors does not significantly improve performance. The optimal SC predictors were cosine, flow graph, and navigation length for FC-Neurosynth; cosine and Euclidean distance for FC-NeuroQuery; and cosine, Euclidean distance, and flow graph for FC-rsfMRI (Fig. 2e). These optimal SC metrics for each FC type were then used to assess SF correspondence by parcel (Fig. 3).

## Multiple linear regression model to compute structure-function correspondence by parcel

To compute SF correspondence by parcel, we used adjusted $R^2$ values produced from a multiple linear regression model for each FC type with the optimal SC measures selected in Fig. 2e. Algebraically, each of these SC measures by parcel are vectors of length 696 and quantify features emanating from a specific YBA parcel to all other 695 YBA parcels. Note that all self-edges (i.e., loops) were not considered in our study. This by-parcel analysis yielded $R^2$ values for each of the 696 parcels. We conducted this analysis separately for FC-Neurosynth, FC-NeuroQuery, and FC-rsfMRI.

## Word embeddings of functional terms

In NLP, a word embedding is a vector representation of a word according to its meaning. The word embeddings themselves measure semantics, and pairwise distances between word embeddings can capture semantic differences, i.e., words that are closer to one another in vector space generally have more similar meaning[23]. To learn vector representations of the selected 334 functional terms, we trained our embedding model on a corpus of 12,909 fMRI papers, the subset of 14,371 total fMRI papers referenced in Neurosynth that had abstracts we could web-scrape. Because the Neurosynth data repository does not contain the explicit texts of the papers, we manually web-scraped the fMRI papers ourselves. Our corpus only consisted of abstracts, as formatting discrepancies and firewalls complicated scraping additional aspects of the papers. We used the *word2vec* R package with 150 dimensions and 20 training iterations to compute the word embeddings using the word2vec technique[52]. Finally, we used tSNE and PCA to map the word embeddings of our 334 functional terms in two representative dimensions, i.e., semantic space.

## Structure-function correspondence by specific function

For each functional term, we determined which parcels were active (i.e., non-zero) and which were inactive (i.e., zero) in the Neurosynth functional activation data. From these active and inactive parcels, we considered their connections. For each term, we computed a vector containing the WM streamline counts (i.e., SC-Count) between all active-active parcel pairs and another vector containing the WM streamline counts between all active-inactive parcel pairs. To reduce the likelihood of having spurious results for some terms due to low numbers of active parcels, we removed functional terms that had less

than 50 active-active connections. This reduced the number of functional terms to analyze from 334 to 313. This analysis was similarly performed for NeuroQuery (Fig. S7c, d). Moreover, this analysis was repeated using SC-Cosine (Fig. S7a,d) rather than SC-Count.

We then executed a Wilcoxon rank-signed test to compare these two vectors for each term. The first vector contains the volume-normalized WM streamline counts between any two active parcels. The second vector contains the volume-normalized WM streamline counts between any one active parcel and one inactive parcel. To quantify SF correspondence, we considered the resulting fold change and $p$ values from this Wilcoxon test. A high fold change value and low $p$ value indicate that parcels which share functional activation for a specific term have a significantly greater number of WM streamlines on average compared to their connections to other parcels inactive for that term. Fold change was computed by dividing the average volume-normalized WM streamline count for active-active connections by that for active-inactive connections for each functional term.

We applied significance thresholds at Bonferroni-corrected $p = 0.05$ and $\log_2(\text{Fold Change}) \approx 0.94$, reflecting the 20th percentile cutoff among significant terms. These significance thresholds separate high SF and low SF terms. Moreover, as an alternative to binarizing SF correspondence, we statistically reevaluated and reaffirmed all correlates using the fold change values as a continuous measure of SF correspondence (Fig. S8).

The correlates of SF correspondence by term that we explored included functional term embeddings and sensory-motor function. For functional term embeddings, we computed the word embeddings for the functional terms and subjected the 150-dimensional space to dimensionality reduction via tSNE or PCA. For sensory-motor function, we sourced mean concreteness ratings from a study by Brysbaert et al. which tasked participants to score 40,000 English words according to involvement in sensory-motor responses[22]. A higher rating indicates more concrete words; a lower rating indicates more abstract words. Eight terms not featured in the concreteness ratings dictionary were omitted from our analysis.

Lastly, for prediction-centric analysis to evaluate how well semantics of functional terms can predict their SF correspondence, the 150-dimensional word embeddings of the functional terms were inputted into either a support vector machine (SVM) binary classifier or a fully connected neural network for regression. For binary classification, the SVM classified functional terms as either high SF or low SF. We implemented the SVM with a linear kernel configured to output probability estimates using *scikit-learn* in Python. We performed four-fold cross-validation and reported the mean AUC and mean accuracy across all folds. We determined four was the maximum number of cross-validation folds where each fold still had a sufficient number of functional terms for statistical power. For the regression analysis, the fully connected feedforward neural network was implemented using *pytorch* in Python. The simple model consists of three linear layers designed for regression tasks. It accepts the 150-dimensional word embeddings as input, processes it through two hidden layers with 128 units each using ReLU activations, and outputs a single value (i.e., fold change as a measure of SF correspondence by term) through the final layer. We similarly performed four-fold cross-validation and reported the mean out-of-sample $R^2$ value across the cross-validation folds, Spearman $r$ value, and $p$ value.

## Macroscale functional gradients that correlate with structure-function correspondence

To examine the correlation across parcels between each of the seven macroscale functional gradients (defined below) and SF correspondence, we computed $R^2$ values and Spearman $r$ correlation coefficient values. The rank-based Spearman approach was chosen to avoid requiring normally-distributed variables and to align with a prior study for comparison[10]. For each correlation, we reported the $R^2$ value,

Spearman correlation coefficient $r$ value, and null model $p$ value. The null model test was conducted by randomly assigning computed functional metrics to parcels over 10,000 runs (Fig. S9).

First, we analyzed the first eigenvector from the diffusion map generated for FC-Neurosynth. Second, we evaluated the first principal component resulting from PCA conducted on FA-Neurosynth. Third, we computed a functional diversity metric to capture the previously hypothesized unimodal-to-transmodal gradient[10]. Functional diversity was calculated as the weighted mean Euclidean distance between word embeddings of the top ten activated functional terms and their corresponding centroid for each parcel. For this and all subsequent functional gradients, the weight applied was the activation score of that term for each parcel in Neurosynth. We only considered the top ten of activated functional terms per parcel to reduce the effect of small type I errors produced from imperfect topic analysis in Neurosynth. The mean number of activated functional terms per parcel was ~55, so taking the top 10 terms reflects the top ~20% of functional terms. We found this amount optimized correlative performance, suggesting some ideal representation of function in each parcel. Note, however, that correlative significance was still observed for relationships when all activated terms were considered. Fourth, the sensory-motor function gradient was measured as the weighted mean concreteness rating of the top ten activated functional terms for each parcel[22]. For the fifth to seventh functional gradients, we referenced the Linguistic Inquiry and Word Count (LIWC2015) dictionary by Pennebaker et al. which assigns words to different psychometric categories[24]. We were particularly interested in the categories of cognitive (e.g., "recall"), perceptual (e.g., "look"), and biological (e.g., "eat"). Note we also evaluated the LIWC categories of affective (e.g., "happy"), social (e.g., "talk"), and drives (e.g., "reward") but did not find any significant correlations with SF correspondence after applying Bonferroni corrections to $p$ values. Moreover, the relativity category (e.g., "go") was not further considered because of its redundant emphasis on motion like our sensory-motor function gradient. Altogether, we mapped the weighted percentage of the top ten activated functional terms per parcel that fall into the LIWC2015 psychometric categories of cognitive, perceptual, and biological[24].

## Cortical thickness

Non-skull stripped T1-weighted images of 200 healthy young adults (100 females, 100 males; mean age 28.85 years; age range from 23 to 36 years) from the HCP were used to derive cortical thickness data. These 200 individuals were a representative subset of the 1065 subjects referenced for WM tractography data; a subset was selected because this number was considered to be sufficient for the analysis reported. The Advanced Normalization Tools (ANTs) cortical thickness pipeline was used to generate cortical thickness values for the gray matter of each T1-weighted image. These results were then nonlinearly normalized to MNI152 space. The cortical thickness values of the voxels making up each Yale Brain Atlas parcel were averaged to produce a single cortical thickness value per parcel. With the cortical thickness values per parcel for 200 subjects acquired, an averaged cortical thickness map was generated and used for correlation tests with SF correspondence and macroscale functional gradients. All analysis of cortical thickness in Yale Brain Atlas space used the 690-parcel-version, which does not feature the six parcels in the corpus callosum.

## Reporting summary

Further information on research design is available in the Nature Portfolio Reporting Summary linked to this article.

## Data availability

The raw dMRI data and processed dMRI data in MNI space can be accessed at https://db.humanconnectome.org/ and https://brain.labsolver.org/hcp_ya.html, respectively, under restricted access to

preserve subject privacy. Access to HCP can obtained by account creation and signing a data use agreement described at https://www.humanconnectome.org/study/hcp-young-adult/data-use-terms. The raw Neurosynth data can be accessed at https://github.com/neurosynth/neurosynth-data. LIWC2015 data is not publicly available due to licensing reasons; users can ascertain access details at https://www.liwc.app/. Except for LIWC2015, the Yale Brain Atlas data and all raw and processed data to replicate methods and results of this paper are available at https://github.com/evancollins1/brain_structure_function. Yale Brain Atlas can be interactively viewed at https://yalebrainatlas.github.io/YaleBrainAtlas/. Source data used to generate figures are provided in the Source data file. Source data are provided with this paper.

## Code availability

All data processing and analysis code is available on our GitHub repository[53]: https://github.com/evancollins1/brain_structure_function.

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

## Acknowledgements

This work was supported by grants from NIH R01 NS109062 (D.D.S. and H.P.Z.) and the Swebilius Foundation (H.P.Z.).

## Author contributions

E.C. conceived the project, with insight from O.C., S.O., D.D.S., and H.P.Z. E.C. processed fMRI from Neurosynth and NeuroQuery. D.D.S. and S.O. aided in simplifying the functional terms of Neurosynth. O.C. processed structural data. O.C. and S.O. developed methods for structural data. A.K. processed cortical thickness data. X.S. and J.A. processed rsfMRI data. H.M. prepared the brain atlas, and E.C. developed software to incorporate the atlas into this project. E.C. led the studies, developed methods, and conducted computational analyses for comparing structural, functional, and cortical thickness data. E.C. interpreted and visualized results. O.C. helped with visualization. E.C. wrote the manuscript. H.P.Z., D.D.S., R.T.C. and X.P. supervised the research. E.C., O.C., S.O., D.D.S. and H.P.Z. revised the manuscript.

## Competing interests
X.P. is a consultant for the Brain Electrophysiology Laboratory Company. X.P. also consults and has a small ownership stake in Veridatand and TytoNyx, which are both small technology startup companies. The remaining authors have no competing interests to declare.
