## [Peer Review File · Nature Communications]

Mapping the structure-function relationship along macroscale gradients in the human brainReviewer #1 (Remarks to the Author):

This draft aims at mapping the correspondence between structure and function in the human brain. Specifically, it relates the diffusion MRI-derived structural connectivity (SC) with the functional MRI-derived functional connectivity (FC). Furthermore, the authors also study meta-analytical functional connectivity (see, e.g., Toro and Burnod 2004, Abdallah et al. 2022), under the name Parcelsynth connectivity, with the SC and FC, and they map the relationship between these connectivities and the macroscale gradients proposed by Margulies et al. (ref 13 in the manuscript).

The goals of the manuscript are ambitious as these relationships have been an open question for decades, addressed in terms of MRI in humans by Honey et al. in 2009 (ref 11 in the manuscript), amongst others. The authors of this manuscript aim to address two specific vital points in the area of structure-function relationships: 1. the issue of underpowered small samples and 2. Incorporating task fMRI into the structure-function relationships.

The aims of this study are of the utmost value to the human neuroscience community, and the study uses modern takes on analyses to perform their work. Still, the work is largely improvable.

The study divides its contributions into three parts that I will analyze separately.

1. Determining representative structural and functional connectivity

The authors have used the HCP1200 dataset, precisely 1,065 subjects, to compute structural connectivity at a region level. They simultaneously used the length of white matter paths and the count as multivariate SC quantitative measurements. The authors also chose to include the Euclidean distance between regions. Suppose this measurement aims to quantify the corticocortical distance between regions. In that case, it is difficult to justify as it will not consider the gyral geometry and will be particularly inaccurate across hemispheres. Using a geodesic distance measurement, such as the one reviewed by Paquola et al. (ref 18 in the manuscript), would have been much more appropriate and easily understandable regarding neuroanatomy.

Then, the authors extract resting-state fMRI-based connectivity and the co-reporting of functional activations in the meta-analytic database Neurosynth. This type of connectivity, namely meta-analytic connectivity, was first analyzed by Toro and Burnod (2004), and it has been a topic of continuous work (see, e.g., Yeo et al. 2015, Abdallah et al. 2022).

2. Structural and Functional Connectivities Demonstrate Little Global Correspondence

In this section, through linear modeling, the authors show that the correlation between structural and functional (either image-based or meta-analytic) is relatively low, around $r=0.3$ in magnitude. The p-values of these correlations are extremely small, ranging close to 10^{-16} , which might be a sign of an overpowered analysis. The manuscript will improve if the authors decide to perform a predictive experiment. That is, to perform a cross-validation experiment where they fit their linear model and subsequently predict functional connectivity from structural connectivity in a left-out set of subjects. Recent studies, such as Varoquaux (2018) and Poldrack and Varoquaux (2020), provide evidence that given that the sample is large enough, prediction-centric analyses are a much better depiction of the accuracy of the correlations studied in this manuscript.

3. Structure-Function Correspondence is higher in primary sensory and motor cortex and lower in association cortex, and changes with task activation

The authors move to analyze the structure-function relationship in specific brain areas, as determined by the principal gradients of FC communicated by Margulies et al. Albeit their argument of finding a structure-function correspondence aligning with the sensory-fugal cortex, I find the evidence in figure 3 challenging to parse. Moreover, the manuscript would benefit from an in-depth analysis of the differences between resting state fMRI and meta-analytic connectivity (or their new name parcelsynth-derived FC). Notably, the meta-analytic connectivity not only includes data from a wide range of fMRI paradigms but also shows information on the agreement of these

studies. Hence, meta-analytic and functional connectivity represent different takes on brain co-activation (e.g., Yeo et al. and Abdallah et al.).

4. Structure-Function correspondence is higher for perceptual and motor functions

The authors claim that perception and motor functions, i.e., non-association functions, exhibit more correspondence. This is in line with studies in animal models, such as Barttfeld et al. (2015), which have shown that the correlation between structure and function can be more significant under anesthesia, pointing to the evidence that primary function is more likely to exhibit a correlation between structure and function. In short, there is converging evidence suggesting the claim of this section.

Nonetheless, the authors' evidence for this claim can be improved. The study performed in this manuscript section through meta-analyses and word embeddings is really interesting. Nonetheless, the correlations reported might also result from an overpowered analysis, for instance, as can be observed in Figure 5D. Performing a prediction-centric analysis, as I have suggested before, will make the evidence brought forward by the authors much more solid. Furthermore, the authors claimed in their previous section, and the introduction that correspondence should be performed in task-based cases on top of resting-state. I fully agree with them and I urge them to push this argument further and perform these analyses on the task of fMRI acquisitions of the HCP dataset. This result should complement their meta-analytic analysis and make their point stronger.

5. Organization of cortical thickness relates to structure-function correspondence

This section addresses an important question, and the study presented is sound. I fail to understand, however, why the authors preformed this analysis only on 100 of the 1,200 subjects of the HCP dataset, while having processed 1,065 for their first claim. Again, the correlations shown in Figures 6B and G, for instance, seem to be more of a product of an overpowered analysis than of the studied phenomena.

In all, the study presented here addresses an important question and from a novel angle. With an improved set of analyses, the evidence produced by the data should make clearer points and render this study very convincing.

Reviewer #2 (Remarks to the Author):

This data-driven study aims to advance knowledge on the human brain's large-scale spatial gradients of structure-function (SF) correspondence. Results support previous hypotheses suggesting that SF is higher in sensorimotor regions and reduced in high-order associative regions. A major novelty of the study is using a meta-analytic approach to infer regional task-based activations and natural language processing (NLP) approaches to compute the most generalizable, representative map of the varying relationship between structure and function. Overall, the paper is well-written, and the aims, analyses and results are clear. I have several questions/comments:

General comments:

1. There may be some level of circularity in the reasoning and the analyses. For example, the Parcelsynth-derived FC likely correlated with cortical thickness, embedding the associations with functional diversity (transitivity). Could the authors please reassure the reviewer that the various analyses are independent?
2. Somewhat related to point 1 above, if rsfMRI-derived FC and Parcelsynth-derived FC are correlated, the results of the new analyses would largely be predicted by what was previously observed in the resting state. Clarifying this point is likely important to highlight the novelty of the current findings.

Technical questions/suggestions/details (in no particular order of importance):

- a. Do the results change with a different SC metric threshold (e.g., paths present in at least 50%

of individuals)? This confirmatory analysis could also provide some preliminary insights into the importance of common and idiosyncratic pathways for SF relations (notwithstanding the lack of individual functional data).

b. Neurosynth may not be considered state-of-the-art by some, and its use may be criticised (as mentioned by the authors in the Discussion). To mitigate this limitation, the authors could assess the relationship between the Neurosynth-derived map and a map derived using the meta-analytic tool Neuroquery.org (<https://elifesciences.org/articles/53385>).

c. SF link has been suggested to relate to neural noise variability across brain regions (e.g. Hearne et al., Mol Psych, 2021). Authors may consider assessing the link between functional diversity and resting-state fMRI signal variability. A link between these two maps may support and add to the notion that increased neural signal variability (within and between regions) relates to reduced SF association. This is a suggestion that the authors may discard.

d. Discussion, 7th paragraph, p. 19. One complementary explanation could be that compared to high-order regions, sensory and motor regions have a low dynamic range (associated with a fast neural timescale that accommodates the quick transition in the response amplitudes of that region when the excitability is increased). The T1w:T2w MRI contrast (sensitive to myelination) is inversely related to neural dynamic range (Pang et al., eLife, 2022), suggesting that the low dynamic range is related to highly myelinated fibres. Authors are free to reject this suggestion should they disagree.

e. L.162: an introduction of spectral clustering and diffusion mapping of parcels would be useful to the reader not familiar with these techniques (or at least refer to methods section).

f. L.200-202: "(...) we found there were significant relationships with FC degree, suggesting parcels that were functionally similar to many other parcels generally had lower SF correspondence", would need to specify that this is true only with parcelsynth, as Fig S2 shows a positive relationship with FC degree for rs-fMRI.

Note that the caption of Fig S2 between (B) and (C) does not match axis labels.

g. L.236: an introduction to what fold change is would be beneficial.

h. L.336: I suggest to be consistent in the use of Spearman r/ρ .

i. L.346: maybe re-mention in the figure caption that FA is functional activation, the reader may spontaneously think fractional anisotropy.

j. L.445-446: "We found functional terms with high SF correspondence had fundamentally different semantic meanings, as measured by a representative axis for word embeddings", isn't it the opposite?

k. L.632-633: "search information" appears twice.

Reviewer #3 (Remarks to the Author):

I co-reviewed this manuscript with one of the reviewers who provided the listed reports as part of the Nature Communications initiative to facilitate training in peer review and appropriate recognition for co-reviewers.

Reviewer #1:

1.0 Our Overall Response

We appreciate the valuable feedback. We believe the insightful comments and well-cited suggestions have significantly enriched the quality of our manuscript. Shown below are our responses to each comment.

1.1 Reviewer #1 Point #1

1. Determining representative structural and functional connectivity

The authors have used the HCP1200 dataset, precisely 1,065 subjects, to compute structural connectivity at a region level. They simultaneously used the length of white matter paths and the count as multivariate SC quantitative measurements. The authors also chose to include the Euclidean distance between regions. Suppose this measurement aims to quantify the corticocortical distance between regions. In that case, it is difficult to justify as it will not consider the gyral geometry and will be particularly inaccurate across hemispheres. Using a geodesic distance measurement, such as the one reviewed by Paquola et al. (ref 18 in the manuscript), would have been much more appropriate and easily understandable regarding neuroanatomy.

Then, the authors extract resting-state fMRI-based connectivity and the co-reporting of functional activations in the meta-analytic database Neurosynth. This type of connectivity, namely meta-analytic connectivity, was first analyzed by Toro and Burnod (2004), and it has been a topic of continuous work (see, e.g., Yeo et al. 2015, Abdallah et al. 2022).

1.1 Our Response:

- The reviewer correctly points out that our manuscript referenced a subset of 1,065 subjects from the HCP1200 dataset. From our original submission, we agree that improvements could be made to better clarify how we selected this subset of 1,065 subjects.
 - To clarify additional details about the methods underlying the selection of this subset, we included more information in lines 576-580 of Methods under the subsection *Structural connectivity*. In short, we obtained these 1,065 processed tractograms from Yeh (*Nat. Commun*, 2022) who initially sourced them from the HCP.

- Second, we appreciate the suggestion of including geodesic distance as a SC metric. The reviewer also raises important concerns regarding Euclidean distance. In the original submission, we chose the simple metric of Euclidean distance due to prior studies in this area: (1) Vázquez-Rodríguez et al. (*PNAS*, 2019) used Euclidean – not geodesic - distance as a SC predictor for FC, and Zamani Esfahlani et al. (*Nat. Commun*, 2022) found that Euclidean distance was the optimal SC predictor for the greatest proportion of brain regions. As for geodesic distance, indeed, the reviewer correctly points out that Paquola et al. (*Trends Cogn. Sci.*, 2022) discusses the work of Margulies et al. (*PNAS*, 2016), which uses geodesic distance to compute the shortest path between any two nodes on the cortical surface. However, we initially decided to not use a conventional geodesic distance metric for two reasons. Firstly, our brain atlas – Yale Brain Atlas – contains parcels in deep structures (i.e., amygdala and hippocampus) not on the cortical surface; thus, the conventional surface-based approaches to geodesic distance would be inappropriate for our volumetric parcellation. Secondly, we preferred to conduct structure-function analysis using whole-brain connectivities, rather than using connectivity based on single hemispheres separately. This preference was due to methodological simplicity and the finding from Zamani Esfahlani et al. (*Nat. Commun*, 2022) showing that structure-function relationships were similar when produced from whole-brain analysis compared to single-hemisphere analyses. With our preference for whole-brain analysis, we referenced Steiner et al. (*Front. Neurosci.*, 2022), which details how “it is not possible to calculate the geodesic distance between the regions that are not in the same hemisphere”. Hence, for these two reasons, we opted not to include a geodesic distance metric similar to that of Margulies et al. (*PNAS*, 2016). Nevertheless, the reviewer’s comment emphasizes an important consideration – to include SC metrics, which like geodesic distance, better capture network information corresponding to neuroanatomical organization. Thus, towards this effort, we have made the following changes.

- To enable more accurate measures of interhemispheric connections for our new SC metrics described below, we have modified the Yale Brain Atlas to include 6 parcels in the corpus callosum. Thus, our atlas has increased from 690 parcels to 696 parcels. We now use this 696-parcel atlas for all analyses except the that of the cortical thickness section, as the corpus callosum is not part of the cortex and thus omitted. For all sections, this atlas change is noted, typically by stating how there is either 696 or 690 parcels considered. Additional details about the 696-

parcel version and 690-parcel version can now be found in lines 565-572 of Methods subsection *Atlas*. With this change in the atlas, all results/figures of the manuscript have been updated except where noted.

- In our first submission, we computed SF correspondence by parcel through a multiple linear regression model with three SC predictors: number of WM streamlines, WM length, and Euclidean distance. Now in our revised work, in order to more comprehensively assess optimal SC predictors that better capture the brain's network organization, we assess 13 different SC metrics, 8 of which are adapted from Zamani Esfahlani et al. (*Nat. Commun*, 2022). These new SC metrics are described in lines 108-118 of Results and lines 599-621 of Methods under the subsection *Structural connectivity*, and illustrated in our updated Figure 1 (see below, specifically Figure 1C).

- We now perform pairwise linear regression models between the 13 SC measures and 3 FC measures (see **Response 2.4** for context on the additional FC type of FC-NeuroQuery) and perform comprehensive model selection via exhaustive search in order to select the optimal SC predictors for each FC type. This is described in lines 167-190 of Results and lines 724-747 of Methods, and illustrated in our updated Figure 2D-E. We ultimately determine cosine, flow graph, navigation length were the optimal predictors for FC-Neurosynth; cosine and Euclidean distance for FC-NeuroQuery; and cosine, Euclidean distance, and flow graph for FC-rsfMRI. Although this comment raises the important consideration that Euclidean distance may not reflect brain geometry, our study

joins Vázquez-Rodríguez et al. (*PNAS*, 2019) and Zamani Esfahlani et al. (*Nat. Commun*, 2022) in finding Euclidean distance as a relatively robust predictor of FC. For these reasons, we opted to continue to consider Euclidean distance in our models.

1.2 Reviewer #1 Point #2

2. Structural and Functional Connectivities Demonstrate Little Global Correspondence

In this section, through linear modeling, the authors show that the correlation between structural and functional (either image-based or meta-analytic) is relatively low, around $r=0.3$ in magnitude. The p-values of these correlations are extremely small, ranging close to 10^{-16} , which might be a sign of an overpowered analysis. The manuscript will improve if the authors decide to perform a predictive experiment. That is, to perform a cross-validation experiment where they fit their linear model and subsequently predict functional connectivity from structural connectivity in a left-out set of subjects. Recent studies, such as Varoquaux (2018) and Poldrack and Varoquaux (2020), provide evidence that given that the sample is large enough, prediction-centric analyses are a much better depiction of the accuracy of the correlations studied in this manuscript.

1.2 Our Response:

- We thank the reviewer for this thoughtful comment, as it has motivated us to comprehensively improve the statistics we report throughout our study. First, as an overall note about the statistical values, we no longer include these very small p-values due to this well-taken concern about their questionable meaning for such high- N analyses. We decided to remove these p-values for high- N association tests across the manuscript because they have minimal interpretable value. One exception where we

maintain reporting p-values is for Figure 4, which we discuss in more detail in **Response 1.4**. Moreover, we now report R^2 values for every association we test, in addition to the Spearman r values as was the case for our original submission. R^2 values provide clear interpretable value, i.e., goodness of fit.

- Second, as for prediction-centric analysis, we have implemented 10-fold cross-validation to assess predictive performance both for the pairwise linear regression models as well as for exhaustively searching multiple linear regression models. These are reflected in the updated Figure 2D-E and Figure S1 (see below).

- Our revised manuscript details these changes to better align with prediction-centric analysis in lines 167-190 of Results and lines 724-747 of Methods under the subsection *Linear regression model selection*.
- Specifically regarding how the cross-validation folds were constructed, note that implementing left-out set of subjects for our analysis as suggested is not suitable. This is because our meta-analytic FC data is at the group-level only. For typical machine learning implementations, the dataset contains both predictor variables and a dependent variable that are individual-specific. Thus, train/test splitting by individual is sensible. However, in our case, only the SC data is individual-specific. For our meta-analytic FC, it is a single group-level representative map, and FC-rsfMRI has been aggregated from a different set of subjects than what is used for SC. Thus, train/test splitting by individual for our analysis is not ideal. If we did split SC data by subject, the train and test partitions would ultimately try to predict the same dependent variable data (i.e., an averaged FC map); therefore, the model's performance on the test set simply would be a proxy measure of how similar the train SC data and test SC data are. Nevertheless, we still recognize

the value of performing prediction with cross-validation; thus, our cross-validation folds were generated using parcel-wise (i.e., node-based) splitting to ensure the test set includes prediction for completely unseen relationships. This detail about our cross-validation splitting is described in lines 729-731 of Methods under *Linear regression model selection*.

1.3 Reviewer #1 Point #3

3. Structure-Function Correspondence is higher in primary sensory and motor cortex and lower in association cortex, and changes with task activation

The authors move to analyze the structure-function relationship in specific brain areas, as determined by the principal gradients of FC communicated by Margulies et al. Albeit their argument of finding a structure-function correspondence aligning with the sensory-fugal cortex, I find the evidence in figure 3 challenging to parse. Moreover, the manuscript would benefit from an in-depth analysis of the differences between resting state fMRI and meta-analytic connectivity (or their new name parcelsynth-derived FC). Notably, the meta-analytic connectivity not only includes data from a wide range of fMRI paradigms but also shows information on the agreement of these studies. Hence, meta-analytic and functional connectivity represent different takes on brain co-activation (e.g., Yeo et al. and Abdallah et al.).

1.3 Our Response:

- This comment has motivated us to update Figure 3 to ease reader interpretation. First, it is important to note that we now predict by-parcel SF correspondence using the optimal multiple linear regression models for each FC type as described above in **Response 1.1 & 1.2**. To make Figure 3 (see below) easier to parse, we have implemented the same color gradient for all three FC types. Moreover, to enable more effective comparisons between the SF map produced from these three FC types, we plotted the R^2 z-scores by region. These z-scores are specific to each FC type so enable easy understanding of which brain regions are relatively higher or lower in SF correspondence when using that FC type. Note that the bars in Figure 3D are arranged from top-to-bottom such that the highest SF R^2 z-score for FC-rsfMRI is on the top and the lowest SF R^2 z-score for FC-rsfMRI is on the bottom.

- Importantly, this comment also emphasizes the distinction between meta-analytic connectivity and functional connectivity arising from task-based fMRI.
 - First, we agree this distinction is valuable and discuss in **Response 1.5** in greater detail about individual-level task-based fMRI data.
 - To make this distinction clearer, we recognize our first submission did not clearly define Neurosynth as a *meta-analytic* resource. In this revision, we ensured to frequently use the descriptor “meta-analytic” to better communicate how Neurosynth (and now also NeuroQuery) represent different takes on brain co-activation compared to functional connectivity from task-based experiments.
 - The reviewer correctly identifies that Neurosynth contains data from a wide range of fMRI paradigms (i.e., resting state and various tasks). We emphasize this in lines 125-126, 236-237 of Results and lines 630-631 of Methods under *Neurosynth-derived functional connectivity*. To isolate activation data from *functions*, we selected a subset of terms in Neurosynth that pertain to engaged

functions (e.g., “write”). Hence, we ultimately term the connectivity data arising from this Neurosynth subset as Neurosynth *functional* connectivity (FC-Neurosynth). In both the original submission and our revision, we selected this subset of functional terms in order to better isolate the effects of explicit functions. With this approach, we hypothesized (although not reported in the manuscript) a larger proportion of the articles contributing to activation were more likely to use task-based fMRI rather than rsfMRI, although we were limited in our ability to concretely quantify this.

- Altogether, our revision more clearly distinguishes between meta-analytic and task-based data. As we discuss in **Response 1.5**, we motivate future studies to explore individual-level task-based fMRI data. The focus of our paper remains devoted to comparing rsfMRI and meta-analytic tools, which, as we discuss in **Response 2.4**, now includes both Neurosynth and NeuroQuery.

1.4 Reviewer #1 Point #4

4. Structure-Function correspondence is higher for perceptual and motor functions

The authors claim that perception and motor functions, i.e., non-association functions, exhibit more correspondence. This is in line with studies in animal models, such as Barttfeld et al. (2015), which have shown that the correlation between structure and function can be more significant under anesthesia, pointing to the evidence that primary function is more likely to exhibit a correlation between structure and function. In short, there is converging evidence suggesting the claim of this section.

Nonetheless, the authors' evidence for this claim can be improved. The study performed in this manuscript section through meta-analyses and word embeddings is really interesting. Nonetheless, the correlations reported might also result from an overpowered analysis, for instance, as can be observed in Figure 5D. Performing a prediction-centric analysis, as I have suggested before, will make the evidence brought forward by the authors much more solid.

1.4 Our Response:

- We thank the reviewer once more for this helpful feedback once again. This comment has motivated us to bolster our findings in this section. We agree with the comment that this analysis could benefit from prediction-centric analysis. As a preliminary note, this comment highlights the small p-value of Figure 4D as overpowered analysis; however,

considering the N is on the order of magnitude of 10^2 , the p-values emerging from the Wilcoxon tests (now $2.92e-13$ and $2.11e-11$) are likely not overpowered for Figure 4D. For this reason, we have decided to continue to report these two p-values in Figure 4D, as it is fundamental to the main takeaway of this section, i.e., there is a significant fundamental difference in semantics between high SF and low SF functional terms.

- Nevertheless, the reviewer's comment more importantly emphasizes the need for prediction-centric analysis. To do accomplish this in this section, we implemented two machine learning models to evaluate how well semantics of functional terms can predict their SC correspondence. As we describe in lines 812-825 of Methods under *Structure-function correspondence by specific function*, "the 150-dimensional word embeddings of the functional terms were inputted into either a support vector machine (SVM) binary classifier or a fully connected neural network for regression. As for binary classification, the SVM classified functional terms as either high SF or low SF. We implemented the SVM with a linear kernel configured to output probability estimates using *scikit-learn* in Python. We performed 4-fold cross-validation and reported the mean AUC and mean accuracy across all folds. We determined four was the maximum number of cross-validation folds where each fold still had a sufficient number of functional terms for statistical power. As for regression, the fully connected feedforward neural network was implemented using *pytorch* in Python. The simple model consists of three linear layers designed for regression tasks. It accepts the 150-dimensional word embeddings as input, processes it through two hidden layers with 128 units each using ReLU activations, and outputs a single value (i.e., fold change as a measure of SF correspondence by term) through the final layer. We similarly performed 4-fold cross-validation and reported the mean out-of-sample R^2 value across the cross-validation folds, Spearman r , and p value." All such changes can be seen in the updated Figure 4 (see below)

- Note that our prediction-centric analysis has replaced the psychometric categorization subfigure that used displayed as Figure 4F in our original submission. We decided to remove our old Figure 4F to minimize redundancy with Figure 5, as it too evaluates the same psychometric categories.

1.5 Reviewer #1 Point #5

... Furthermore, the authors claimed in their previous section, and the introduction that correspondence should be performed in task-based cases on top of resting-state. I fully agree with them and I urge them to push this argument further and perform these analyses on the task of fMRI acquisitions of the HCP dataset. This result should complement their meta-analytic analysis and make their point stronger.

1.5 Our Response:

- This comment importantly highlights the advantages of supplementing our findings by including task-based data. We entirely agree with this perspective and emphasize this point in lines 472-473, 558 of Discussion for future research. There were a few reasons we ultimately decided not to embark on this effort to include task-based data in the revised manuscript.

- First, task-based analysis for each of the 7 tasks from the HCP would require individual-level analysis that would require a significant amount of computational processing time. This is principally because the group-level task-based data included in HCP is registered at the surface level in a space called MSMAll, not in the MNI152 volumetric space that we use throughout this manuscript. Thus, as we clarified with the HCP team in this forum, incorporating the HCP task-based data would require extensive individual-level analysis.
- Our principal focus for this paper is to compare meta-analytic functional data with resting state functional data. Thus, we reasoned that to additionally include task-based data was outside the scope of this study, and that this would potentially distract readers from the core takeaways sufficiently conveyed by contrasting meta-analytic data with resting-state data. Moreover, we believe a comprehensive, individual-level analysis of task-based fMRI to assess SF correspondence could warrant a stand-alone future publication.
- Considering these reasons, we instead devoted efforts to include another meta-analytic data source, i.e., NeuroQuery. See **Response 2.4** for more information.

1.6 Reviewer #1 Point #6

5. Organization of cortical thickness relates to structure-function correspondence

This section addresses an important question, and the study presented is sound. I fail to understand, however, why the authors performed this analysis only on 100 of the 1,200 subjects of the HCP dataset, while having processed 1,065 for their first claim. Again, the correlations shown in Figures 6B and G, for instance, seem to be more of a product of an overpowered analysis than of the studied phenomena.

1.6 Our Response:

- We thank the reviewer for this opportunity to clarify. In our original submission, we included 100 subjects of the total 1,200 due to the limits on computational processing time. Hence, we opted to include 100 subjects, an amount with a sufficient statistical power to ensure representative group-level data. Nevertheless, since our first submission, we have increased the cortical thickness N from 100 to 200. These details are more clearly detailed in lines 862-866 of Methods under the subsection *Cortical thickness*. We note that the results do not change appreciably between $N = 100$ and $N = 200$, suggesting that the number we have included is adequate.

- As for the comment of this analysis being overpowered, we addressed this by omitting these p-values of minimal interpretable value; we instead only report R^2 and Spearman r values in Figure 6 (see below). Moreover, on a related note, reviewer #2 raised an important point regarding multicollinearity; thus, our changes to verbiage in this section also reflect our amendments spelled out in **Response 2.1**.

Reviewer #2:

2.1 Reviewer #2 Point #1

There may be some level of circularity in the reasoning and the analyses. For example, the Parcelsynth-derived FC likely correlated with cortical thickness, embedding the associations with functional diversity (transitivity). Could the authors please reassure the reviewer that the various analyses are independent?

2.1 Our Response:

- The reviewer's comment importantly raises concerns about circularity and statistical independence. We wanted to assure the reviewer the various analyses are indeed independent, and our one concern about multicollinearity in the cortical thickness section has now been explicitly detailed in the revised manuscript.
- To provide more detail, we commonly assessed the relationship between connectivity-based measures and the SF coupling R^2 value, for example, in Figure S5 where we assess the relationship between connectivity degree and SF R^2 . This analysis is precisely analogous to Figure 2C of the important publication by Vázquez-Rodríguez et al. (*PNAS*, 2019).

- Moreover, we commonly assess the relationship with functional activation data and the SF coupling R^2 value, for example, in Figure 5 as we assess macroscale gradients. This analysis from a circularity-perspective is analogous once again to Figure 4B of Vázquez-Rodríguez et al. (*PNAS*, 2019).
- We did, however, make a few changes to better minimize circularity. We appreciate being motivated to make these edits.
 - First, we have removed the original Figure 4F from our first submission. As we detail in **Response 1.4**, our prediction-centric analysis has replaced the psychometric categorization subfigure that used to occupy Figure 4F in our first submission. Although our new prediction-centric analysis is important in its own right, we have decided to omit the original Figure 4F due to concerns about redundancy, as Figure 5 ultimately evaluates the same psychometric categories.
 - As detailed thoroughly in **Response 2.4**, we opt to include NeuroQuery as an additional FC data type. With respect to circularity and independence, we ensure to avoid using FC-NeuroQuery for any of our downstream analysis involving word embeddings, as NeuroQuery's embedding-based generation of activation data would compromise statistical independence. We state this in lines 261-263 of Results and lines 659-661 of Methods.
 - In the effort to be more cognizant of multicollinearity, we now mention the variance inflation factors (VIFs) for the various functional gradients analyzed in Figure 5; we mention this in lines 387-388 of Results.
 - Lastly, also to better address multicollinearity, we have edited our commentary in the Results section for cortical thickness. As you correctly identify, cortical thickness being significantly correlated with both SF coupling and, for example, FC diffusion component (also a correlate with SF coupling) suggests a degree of multicollinearity. Thus, we now explicitly state the following in lines 439-444 of Results: "These correlations that are significant suggest some degree of multicollinearity between cortical thickness and other correlates of SF correspondence. Despite having relatively low VIFs ranging from 1.2 to 3.4, multicollinearity concerns prevent any definitive conclusions about the way in which cortical thickness directly influences SF correspondence; nevertheless, our study reports an association with cortical thickness pertinent to SF coupling and other macroscale functional gradients." We opted to keep these results because, regardless of which factor is *directly* influencing cortical thickness, we consider

our results to be thought-provoking and pertinent to our study. We encourage future studies to better probe the relationship with cortical thickness to establish causality.

2.2 Reviewer #2 Point #2

Somewhat related to point 1 above, if rsfMRI-derived FC and Parcelsynth-derived FC are correlated, the results of the new analyses would largely be predicted by what was previously observed in the resting state. Clarifying this point is likely important to highlight the novelty of the current findings.

2.2 Our Response:

- We thank the reviewer for inviting us to clarify the novelty of our work. The reviewer correctly points out our finding that the spatial patterning of SF coupling across the brain is largely similar when using FC-rsfMRI and using FC-Neurosynth. This is illustrated in Figure 3.
- As we mention in multiple places of our manuscript, the principal novelty of our work is the use of meta-analytic FC to identify SF coupling patterns across hundreds of specific functions (Figure 4). This, for example, is stated in lines 259-260 of Results. To make this point even clearer directly within our Results of Figure 3, we now have included the following in lines 225-228: “This overall alignment mitigated some of our concerns about the noise level from the meta-analytic approaches and motivated us to leverage the main advantage of these meta-analytic repositories, i.e., being able to analyze activity across hundreds of specific functions, in the next section of our study.”
- Lastly, although they are correlated, they have key differences. As we did in our original submission, we address these differences in lines 230-237 of Results.

2.3 Reviewer #2 Point #3

Do the results change with a different SC metric threshold (e.g., paths present in at least 50% of individuals)? This confirmatory analysis could also provide some preliminary insights into the importance of common and idiosyncratic pathways for SF relations (notwithstanding the lack of individual functional data).

2.3 Our Response:

- Thank you for motivating us to perform this analysis. Now, in Figure S2 (see below) we evaluate the effect of zero-thresholding as well as weighting. As we state in lines 589-

597 of Methods under *Structural connectivity*, “We experimented with differing zeroing thresholds (i.e., any connections must have nonzero WM streamlines in more than some X% of subjects in order to not be zeroed in the group-level matrix) and found maximal SF correspondence without any zeroing threshold (Figure S2A). Additionally, we evaluated the effect of weighting SC-Count either by multiplying or dividing by WM length and found maximal SF correspondence without any weighting (Figure S2B). Thus, with our primary interest being maximizing SF correspondence, we maintained the unthresholded, unweighted, volume-normalized SC-Count group-level matrix. Nevertheless, further research is encouraged to explore how thresholding and weighting better capture underlying group-level neuroanatomy.”

- Relatedly, this comment motivated us to consider another way to amend raw SC-Count. As done in Zamani Esfahlani et al. (*Nat. Commun*, 2022), in our revised work, we normalized SC-Count by dividing the averaged streamline count between any two parcels by the geometric mean volume of the two parcels. We do such volume normalization so that SC-Count is not higher for more voluminous parcels that, say for example, have the same density of WM streamlines as some less voluminous parcels. This is specified in lines 587-589 of Methods under *Structural connectivity*.

2.4 Reviewer #2 Point #4

Neurosynth may not be considered state-of-the-art by some, and its use may be criticised (as mentioned by the authors in the Discussion). To mitigate this limitation, the authors could assess the relationship between the Neurosynth-derived map and a map derived using the meta-analytic tool Neuroquery.org(<https://elifesciences.org/articles/53385>).

2.4 Our Response:

- We thank the reviewer for drawing our attention to NeuroQuery. This comment has motivated us to supplement most findings in our work by additionally including FC-NeuroQuery as an alternative meta-analytic-derived FC. Thus, most analyses in our paper now compare across three FC types – rsfMRI, Neurosynth, and NeuroQuery. In addition to affecting our commentary in Results and Discussion, we have included a separate Methods section for *NeuroQuery-derived functional connectivity* in lines 649-661.
- The main figures that were principally changed due to the inclusion of NeuroQuery include Figures 1-3 (see below) as well as Figures S1-5, S7.

- Importantly, though, we highlight in both Results (lines 260-263) and Methods (lines 658-661) under *NeuroQuery-derived functional connectivity*, “For our downstream analyses involving word embeddings, we chose to reference FC-Neurosynth data rather than FC-NeuroQuery to not compromise statistical independence considering NeuroQuery uses word embeddings to generate activation data.” Thus, we still mainly use FC-Neurosynth for Figures 4-6 in the revised manuscript. Using FC-NeuroQuery for Figure 4, for example, would be problematic because we identify word embeddings to successfully predict SF coupling; however, since word embeddings were used to merge activation data in NeuroQuery itself, we would largely be biasing our analyses towards this conclusion already. Thus, we use Neurosynth for most NLP analyses in our revision, as Neurosynth did not use semantic smoothing to generate the activation data.
- As a note on terminology, in our original submission we termed functional activation data from Neurosynth as *Parcelsynth*. In this revision, considering we now have both Neurosynth and NeuroQuery data, for simplicity, we have opted to drop the name *Parcelsynth*. We now simply directly refer to the meta-analytic repository name, i.e., Neurosynth or NeuroQuery.

2.5 Reviewer #2 Point #5

SF link has been suggested to relate to neural noise variability across brain regions (e.g. Hearne et al., Mol Psych, 2021). Authors may consider assessing the link between functional diversity and resting-state fMRI signal variability. A link between these two maps may support and add to the notion that increased neural signal variability (within and between regions) relates to reduced SF association. This is a suggestion that the authors may discard.

2.5 Our Response:

- We thank the reviewer for highlighting the fascinating work of Hearne et al. (*Mol Psych*, 2021), which importantly highlights an association between neural noise level (i.e., variance of explanatory variables) and SF coupling. We agree this analysis could aid a paper focused on SF coupling; however, we ultimately decided not to pursue studying the link between rsfMRI signal variability and functional diversity in our paper. We reasoned that our main focus and novelty of our study stems from the meta-analytic FC data, namely analyzing SF coupling by specific functional term. The rsfMRI-derived FC largely serves as a control for us. Thus, because the analysis suggested by this comment would only be pertinent to our rsfMRI-derived FC (due to us only having subject-level temporal data at resting-state), we decided this would be outside the scope of this study. This analysis, however, could be well-served in another future study of SF coupling tackling subject-level task-based fMRI data.

2.6 Reviewer #2 Point #6

Discussion, 7th paragraph, p. 19. One complementary explanation could be that compared to high-order regions, sensory and motor regions have a low dynamic range (associated with a fast neural timescale that accommodates the quick transition in the response amplitudes of that region when the excitability is increased). The T1w:T2w MRI contrast (sensitive to myelination) is inversely related to neural dynamic range (Pang et al., eLife, 2022), suggesting that the low dynamic range is related to highly myelinated fibres. Authors are free to reject this suggestion should they disagree.

2.6 Our Response:

- We appreciate highlighting the recent work of Pang et al. (*eLife*, 2022). We agree this is a complementary explanation. We have added the following sentence in lines 552-554 of Discussion: “This explanation is complemented by the finding that sensory-motor

regions have lower dynamic range, enabling them to quickly transition between low and high activity levels.” We cite Pang et al. (*eLife*, 2022).

2.7 Reviewer #2 Point #7

L.162: an introduction of spectral clustering and diffusion mapping of parcels would be useful to the reader not familiar with these techniques (or at least refer to methods section).

2.7 Our Response:

- We now refer the reader to see Methods in line 167 of Results when we introduce the results from spectral clustering and diffusion mapping.

2.8 Reviewer #2 Point #8

L.200-202: “(...) we found there were significant relationships with FC degree, suggesting parcels that were functionally similar to many other parcels generally had lower SF correspondence”, would need to specify that this is true only with parcelsynth, as Fig S2 shows a positive relationship with FC degree for rs-fMRI.

Note that the caption of Fig S2 between (B) and (C) does not match axis labels.

2.8 Our Response:

- We thank the reviewer for drawing our attention to this ambiguity. In our revised submission, in lines 243-246 of Results, we now clarify: “...we found there was a significant relationship with FC-Neurosynth, suggesting parcels that were functionally similar to many other parcels in Neurosynth generally had lower SF correspondence. This was not as significantly the case for FC-rsfMRI or FC-NeuroQuery, potentially due to their activation data being less sparse than that of Neurosynth.”
- Secondly, the reviewer is correct about the inaccuracy of the initial caption for Figure S2 (now Figure S5 in our revision); we incidentally swapped the rsfMRI and Parcelsynth descriptions. In our comparable Figure S5 in our revision, we ensure the caption is correctly ordered.

2.9 Reviewer #2 Point #9

L.236: an introduction to what fold change is would be beneficial.

2.9 Our Response:

- We thank the reviewer for pointing this out. From this comparable line 269 in Results, we now direct the reader to see Methods. In lines 791-793 of Methods under subsection *Structure-function correspondence by specific function*, we now specify “Fold change was computed by dividing the average volume-normalized WM streamline count for active-active connections by that for active-inactive connections for each functional term.”

2.10 Reviewer #2 Point #10

L.336: I suggest to be consistent in the use of Spearman r/rho.

2.10 Our Response:

- We thank the reviewer for pointing this out. In our revised manuscript, all instances are consistently Spearman r.

2.11 Reviewer #2 Point #11

L.346: maybe re-mention in the figure caption that FA is functional activation, the reader may spontaneously think fractional anisotropy.

2.11 Our Response:

- We now specify again that functional activation is FA in the caption of Figure 5.

2.12 Reviewer #2 Point #12

L.445-446: “We found functional terms with high SF correspondence had fundamentally different semantic meanings, as measured by a representative axis for word embeddings”, isn’t it the opposite?

2.12 Our Response:

- We thank the reviewer for pointing out this ambiguity. This sentence should have been more clearly stated as follows: “We found functional terms with high SF correspondence had fundamentally different semantic meanings compared to functional terms with low SF...”. In our revision, we have modified the syntax of the comparable sentence to be even clearer; as we state in line 506-507 of Discussion: “We found that semantic embeddings of functional terms could successfully predict their SF correspondence.”

2.13 Reviewer #2 Point #13

L.632-633: “search information” appears twice.

2.13 Our Response:

- We thank the reviewer for pointing this out. In our comparable sentence in our revised manuscript found in line 605-609 of Methods under subsection *Structural connectivity*, we no longer have this redundancy.

Reviewer #3:

As reviewer #3 co-reviewed this manuscript with reviewer #1, please refer to our point-by-point response to reviewer #1. We thank the reviewer for their time and consideration in providing thoughtful feedback.

Reviewer #2 (Remarks to the Author):

The authors have comprehensively addressed our concerns. Congratulations for this thoughtful study.

Reviewer #3 (Remarks to the Author):

I co-reviewed this manuscript with one of the reviewers who provided the listed reports as part of the Nature Communications initiative to facilitate training in peer review and appropriate recognition for co-reviewers.